# REVERSE-ENGINEERED REASONING FOR OPEN-ENDED GENERATION

**Haozhe Wang**[1,2,*], **Haoran Que**[1,3], **Qixin Xu**[6], **Minghao Liu**[4,5], **Wangchunshu Zhou**[4], **Jiazhan Feng**[1]
**Wanjun Zhong**[1], **Wei Ye**[3], **Tong Yang**[3], **Wenhao Huang**[1], [†]**Ge Zhang**[1,4], [†]**Fangzhen Lin**[2]
[1]ByteDance Seed,     [2]Hong Kong University of Science and Technology,
[3]Peking University,     [4]M-A-P,     [5]2077AI,     [6]Tsinghua University
[*]Work done at ByteDance Seed,     [†]Corresponding authors

## ABSTRACT

While the "deep reasoning" paradigm has spurred significant advances in verifiable domains like mathematics, its application to open-ended, creative generation remains a critical challenge. The two dominant methods for instilling reasoning—reinforcement learning (RL) and instruction distillation – falter in this area; RL struggles with the absence of clear reward signals and high-quality reward models, while distillation is prohibitively expensive and capped by the teacher model's capabilities. To overcome these limitations, we introduce REverse-Engineered Reasoning (REER), a new paradigm that fundamentally shifts the approach. Instead of building a reasoning process "forwards" through trial-and-error or imitation, REER works "backwards" from known good solutions to computationally discover the latent, step-by-step deep reasoning process that could have produced them. Using this scalable, gradient-free approach, we curate and open-source DeepWriting-20K, a large-scale dataset of 20,000 deep reasoning trajectories for open-ended tasks. Our model, DeepWriter-8B, trained on this data, not only surpasses strong open-source baselines but also achieves performance competitive with, and at times superior to, leading proprietary models like GPT-4o and Claude 3.5.

## 1 INTRODUCTION

The paradigm of "deep reasoning" is catalyzing a shift in large language model (LLM) reasoning, moving beyond rapid, surface-level inference to leverage increased computational investment at test time (Guo et al., 2025; Jaech et al., 2024; Team, 2025; Muennighoff et al., 2025; Fu et al., 2025). This approach unlocks sophisticated capabilities like multi-step planning and complex problem-solving, yielding remarkable performance gains in verifiable domains such as mathematics and programming. The success in these areas has been largely propelled by Reinforcement Learning (RL), where clear reward signals for correct outcomes effectively guide a model's search through vast solution spaces.

However, the reliance on verifiability presents a formidable barrier when applying deep reasoning to open-ended, creative domains (Lu, 2025; Ouyang et al., 2022). Creative writing, a quintessential example of a non-verifiable task, lacks a singular, objective ground truth. Instead, its quality is judged on subjective criteria like originality, emotional resonance, and narrative coherence (Wu et al., 2025). This disconnect raises a critical and largely unexplored research question:

**How to instill deep reasoning for open-ended generation in the absence of task verifiability?**

Bridging this gap is profoundly challenging. The dominant paradigms for cultivating advanced reasoning falter here; adapting RL by training a reward model to approximate subjective quality that aligns with human preferences is an immense challenge in itself (Ouyang et al., 2022), and the subsequent RL process is notoriously sample-inefficient and computationally burdensome (Lu, 2025). The alternative, instruction distillation from a powerful model, is often prohibitively expensive and fundamentally capped by the teacher model's capabilities (Toshniwal et al., 2024). This is exacerbated by the scarcity of high-quality queries and deep reasoning trajectories tailored for complex open-ended generation (Bai et al., 2024). These constraints create a critical bottleneck, demanding a new paradigm that sidesteps both the sample inefficiency of RL and the costly dependency of distillation.

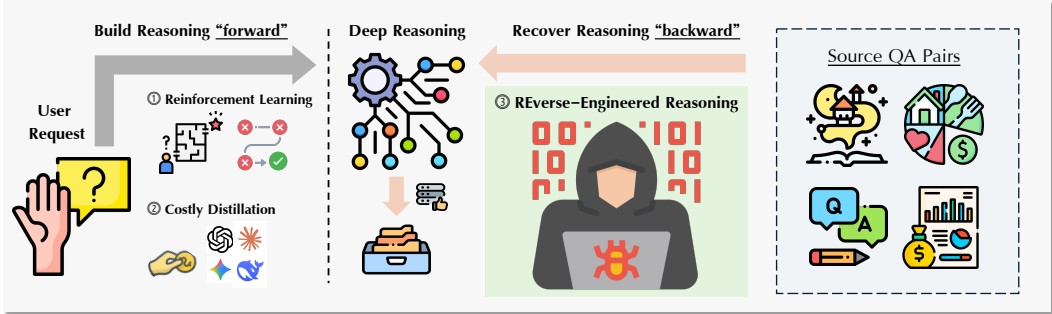

Figure 1: **(Left)** Existing methods attempt to build deep reasoning "forwards" for a user request through trial-and-error (RL) or costly distillation, which falter in open-ended domains that lack clear, verifiable reward signals. **(Right)** We propose a third path for teaching deep reasoning, REverse-Engineered Reasoning (REER). REER works "backwards", recovering plausible human-like thought process from known-good outputs in open-source Question-Answer (QA) pairs.

To break this impasse, we introduce a new paradigm: **REverse-Engineered Reasoning (REER)**. In contrast to conventional methods that build a reasoning process "forwards" through trial-and-error or distillation, *we work "backwards" from a known good outcome*. We essentially ask: "Given a high-quality piece of output, what is the most coherent and logical thinking process that would have generated it?" By answering this question, we can synthesize the otherwise latent, human-like reasoning paths at scale, bypassing costly distillation of thinking data beforehand or inefficient trial-and-error.

We pioneer a novel approach that operationalizes the REER paradigm and, for the first time, *instill deep reasoning capabilities for open-ended generation entirely from scratch*. Our approach involves three key stages. First, we source a diverse dataset of query-solution pairs for open-ended generation from the web, encompassing 16,000 samples spanning across ordinary-life question-answering, academic writing, functional writing and creative writing. From these, we "reverse-engineer" deep reasoning trajectories – structured, human-like thought process tailored for open-ended generation. Eventually, we use this synthetic data to fine-tune a base language model, teaching it to reason and plan deeply before generating a final solution.

The central innovation lies in how we synthesize this data: *we formulate the recovery of high-quality thinking trajectories as a gradient-free search problem.* These trajectories are found by iteratively refining an initial plan, with the search guided by a proxy for thought quality – the perplexity of a known good solution. The gradient-free, self-contained nature of our synthesis process lends us the scalability. By obviating the need for expensive, query-by-query distillation from proprietary models or the sample-inefficiency of reinforcement learning, our approach provides a cost-effective and automatable pathway to generate vast quantities of high-quality, deep-thinking training data. This makes it possible to instill sophisticated reasoning in models at a scale that was previously impractical.

Using this method, we created **DeepWriting-20K**, a comprehensive dataset of 20,000 thinking trajectories, and fine-tuned a Qwen3-8B base model. Our extensive empirical evaluation on benchmarks like LongBench (Bai et al., 2024), HelloBench (Que et al., 2024), and WritingBench (Wu et al., 2025; Kang et al., 2025) shows that DeepWriter-8B successfully internalizes this deep reasoning process. It not only substantially outperforms strong open-source baselines but also achieves performance competitive with, and in some cases exceeding, leading proprietary models like GPT-4o and Claude 3.5, validating our approach as a powerful new pathway for building complex reasoning for open-ended generation.

Our primary contributions are:

- **Pioneering a New Paradigm for Reasoning**: We introduce Reverse-Engineered Reasoning (REER), a novel and scalable paradigm that synthesizes high-quality thinking trajectories without reliance on RL or distillation. This work charts a new, third path for teaching complex reasoning, addressing a critical bottleneck in the field.

- **Enabling New Research with DeepWriting-20K:** We contribute a comprehensive dataset of 20,000 query-response and deep reasoning trajectories across 25 categories. This resource is designed to mitigate data scarcity and catalyze future research into the mechanisms of planning and structured thought in open-ended generation.
- **Strong Performance in Open-Ended Generation from Scratch:** We provide rigorous empirical evidence that our approach works. Our model, DeepWriter-8B, trained entirely on our synthesized data, matches or exceeds the performance of premier proprietary models on challenging writing benchmarks, proving that human-like deep reasoning can be cultivated without costly distillation or RL.

## 2 APPROACH

---

**[Question]**
Write an engaging story based on the following writing prompt:
  "You can't speedrun an Isekai! It's not a game!" The God tells me.
  "Watch me." I reply.
**[Response]**
<think>
Alright, the user wants a story based on the prompt ...
First, I need to understand the user's intent. The core objective is to create a compelling ...
Let me think... Maybe I should emphasize the protagonist's journey from disbelief to ...
Hmm... Alternatively, maybe I should also consider the emotional journey of the protagonist and ...
For the content, I need to brainstorm a core creative idea.
The main character, Hiro ...
Building the structure, I'll use a narrative framework with a clear beginning, development, climax.
Let me think about the beginning ... Next, the development phase ... For the climax, I'll need a final, epic battle ... Additionally, I should include Hiro's reflections on his journey ...
Wait, that's a bit too straightforward.
Let me add some twists and turns.
For instance, the God could ... Hmm, maybe I should include a subplot involving ... This could add layers to the story. ...
</think>
<answer>
...
</answer>

---

Figure 2: An example of the deep reasoning generated by DeepWriter. The model demonstrates deliberate planning for open-ended generation, employing various thinking patterns (e.g., 'Hmm... Alternatively', 'Wait, that's a bit ...') to facilitate structured reasoning, including logical deduction, branching, and backtracking.

Our central goal is to instill deep reasoning in LLMs for open-ended tasks without relying on costly distillation or reinforcement learning. To achieve this, we introduce **REverse-Engineered Reasoning (REER)**, a novel paradigm that shifts the objective from generating a solution to discovering the latent reasoning process behind an existing high-quality one. Instead of building a reasoning process "forwards" via trial-and-error, REER works "backwards" from a known good output to computationally synthesize the step-by-step thinking that could have produced it. This approach is operationalized as a search problem where we iteratively refine an initial thinking process to discover a trajectory that best explains a high-quality, human-written output. An example of the structured reasoning we aim to cultivate is shown in **Figure 2**, where the model demonstrates deliberate planning, exploration of alternatives ("Hmm... Alternatively"), and self-correction ("Wait, that's a bit too straightforward").

### 2.1 REVERSE-ENGINEERED REASONING AS A SEARCH PROBLEM.

Let $x$ be an input query (e.g., a story prompt) and $y$ be a high-quality reference solution (e.g., a well-written story). Our objective is to find a *deep reasoning trajectory*, denoted by $z$, which represents a structured, step-by-step thinking process that guides the generation of $y$ from $x$.

The primary challenge in open-ended domains is the absence of a verifiable correctness signal. The REER paradigm circumvents this by reframing the problem: instead of judging the final output, we evaluate the quality of a *thinking process* based on how well it explains a known-good output. We operationalize this principle by using the **perplexity** (a.k.a., the model surprise) of the reference solution $y$ as a proxy for the quality of a given reasoning trajectory $z$. A lower perplexity score for $y$, conditioned on both $x$ and $z$, indicates that the trajectory provides a more coherent and effective plan. In essence, REER posits that a good thinking process $z$ is one that makes a high-quality answer $y$ seem maximally probable and logical to the model.

Formally, we model the deep reasoning trajectory $z$ as a discrete sequence of reasoning steps, $z = [z_1, z_2, \ldots, z_n]$. The problem is then formulated as a search for the optimal trajectory $z^*$ within the vast space of possible trajectories $\mathcal{Z}$, such that $z^*$ minimizes the perplexity of the reference solution $y$:

$$z^* = \arg\min_{z \in \mathcal{Z}} \text{PPL}(y|x, z)$$

Here, $\text{PPL}(y|x, z)$ is the perplexity of the token sequence of $y$ as calculated by a generator LLM, conditioned on $x$ and $z$. This optimization is performed via a *gradient-free local search algorithm*, allowing us to iteratively refine the trajectory without a differentiable objective.

## 2.2 ITERATIVE REFINEMENT VIA LOCAL SEARCH

Solving for the optimal trajectory $z^*$ directly is intractable due to the vast search space. Therefore, we propose an iterative refinement algorithm that employs a guided local search to discover a high-quality deep reasoning trajectory. The algorithm starts with an initial trajectory and progressively improves it by refining its constituent segments, guided by the perplexity signal. As visualized in **Figure 3**, the algorithm runs as follows:

**1. Initialization:** For a given $(x, y)$ pair, we generate an initial, imperfect deep reasoning trajectory, $z^{(0)}$, by prompting an LLM with a thought-provoking instruction (see Appendix, Listing 1) to produce a plausible plan. This initial trajectory is denoted as $z = [z_1, z_2, \ldots, z_n]$.

**2. Node Expansion (Segment-wise Edits):** The core of our method is an iterative loop that refines $z$ one segment at a time. In each iteration, we select a segment $z_i$ to improve. We prompt the LLM to generate candidate refinements with more thinking-based details, elaborations and reflections. To generate these refinements, we provide the full context including the query $x$, the reference solution $y$, and the surrounding trajectory segments (refined steps $z^*_{<i}$ and initial steps $z_{>i}$). The prompt is meticulously designed to encourage detailed reasoning while preventing the model from simply copying content from the reference solution (see Appendix, Listing 2).

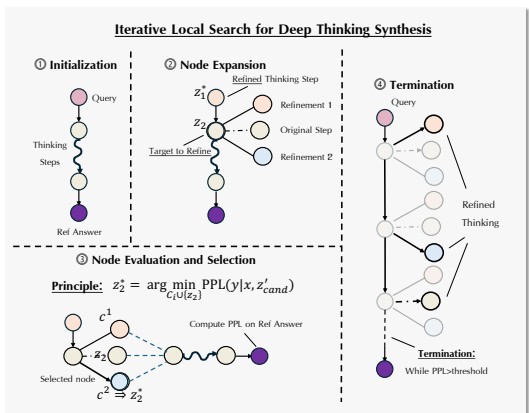

Figure 3: Method Overview: Iterative Local Search for deep reasoning Synthesis.

**3. Node Evaluation and Selection:** For each generated candidate $c$, we construct a temporary trajectory $z'_{\text{cand}}$ by substituting $z_i$ with $c$. We then evaluate each candidate by computing its quality score, $S(c) = \text{PPL}(y|x, z'_{\text{cand}})$. The candidate with the lowest perplexity score is chosen as the updated segment for the next iteration: $z^*_i = \arg\min_{c \in \mathcal{C}_i \cup \{z_i\}} S(c)$. We include the original segment $z_i$ in the candidate set to ensure that the perplexity improves monotonically.

**4. Termination:** The refinement process repeats until the perplexity of the solution reaches a predefined target threshold or a maximum number of iterations is completed. The final output is a refined trajectory $z^*$.

This process allows us to create a dataset of $(x, z^*, y)$ triples, which can then be used to fine-tune a base LLM to internalize the deep reasoning capability for open-ended generation from scratch.

It is important to distinguish our iterative local search from methods like Monte Carlo Tree Search (Browne et al., 2012; Li et al., 2025). First, by using the perplexity of a complete ref-

erence solution as a quality proxy, REER avoids the computationally expensive rollouts required in MCTS. Second, our approach operates on a "global-to-local" principle: we start with a complete, albeit imperfect, global plan and iteratively improve it through local, segment-wise edits. This contrasts with standard MCTS or beam search, which build solutions sequentially by extending partial states. These distinctions make our approach a scalable and efficient method for operationalizing REER, enabling the creation of large-scale, deep-reasoning datasets for open-ended generation.

## 2.3 TRAINING DATA CURATION

The success of our methodology hinges on a large-scale, high-quality dataset of $(x, z^*, y)$ triples. The creation of this dataset follows a multi-stage pipeline: sourcing diverse query-solution pairs, synthesizing deep reasoning trajectories, and applying rigorous filtering.

**2.3.1 Sourcing of (Query, Solution) Pairs.** To ensure diversity in style, topic, and complexity, we sourced initial (x,y) pairs from three primary channels. We gathered prompt-story pairs from online communities like r/WritingPrompts, using upvotes as a quality proxy, and reverse-engineered queries (x) from classic Project Gutenberg texts (y) using GPT-4o. Finally, we augmented this collection with data from instruction tuning datasets such as WildChat (Zhao et al., 2024) and LongWriter6K (Bai et al., 2024).

**2.3.2 Trajectory Synthesis and Filtering.** From our sourced pairs, we selected 20,000 high-quality query-solution pairs covering 25 manually nominated categories to ensure broad topic coverage. For each pair, we executed our iterative local search algorithm to generate an optimal deep reasoning trajectory $z^*$.

**Context Engineering.** The efficacy of the search algorithm, however, hinges not only on the search procedure but also on the nuanced design of the instructions used to elicit deep reasoning from the generator LLM. We proposed three key designs in our context engineering to ensure high-quality synthesis. We only summarize the key insights here and refer the reader to the appendix for detailed prompts.

1. **Enforcing Segment-wise Edits via a Meta-Structure.** To ensure the generator model performs a true segment-wise edit without including edits for the subsequent parts of the trajectory, we enforce a *meta-structure* for the reasoning process within the prompt. This serves as an *implicit regularizer*, helping the model to localize the current segment and constrain its edits to the intended scope when performing segment-wise edits.

2. **Injecting Human-like Thinking Patterns.** To prevent the synthesis of rigid and formulaic reasoning, we *deliberately inject human-like thinking patterns*. Prompts explicitly encourage phrases that signify cognitive exploration and self-reflections (e.g., "Hmm...maybe I can...", "Wait, that's a bit..."), triggering a more human-like reasoning style and incentivizing self-reflection through training (Wang et al., 2025b).

Analysis of this synthesis process, shown in **Figure 4**, confirms its effectiveness. The perplexity distribution shifts significantly lower after refinement, with the vast majority of samples showing a marked PPL improvement. Concurrently, the token length of the trajectories increases, to an indicating that the search process successfully expands initial simple plans into more detailed and elaborate reasoning chains.

During instruction tuning, we witness the challenge of repetitive and degenerate thinking. We therefore applied two heuristic filtering strategies to prune low-quality trajectories:

1. **End-of-Thinking Filtering:** We discarded samples where thinking patterns persisted in the final 10% of the sequence. These trajectories risk misleading the model to stuck in a repetitive loop and failing to conclude its reasoning process.

2. **Repetition Filtering:** We employed a repetition metric to measure the frequency of the top-3 n-grams within each trajectory. Samples exhibiting high n-gram repetition, a sign of degenerative looping expressions, were filtered out.

This process resulted in a final dataset of 20,000 high-quality deep reasoning trajectories. The distribution of this dataset, shown in **Figure 5**, highlights its diversity, with a significant focus on

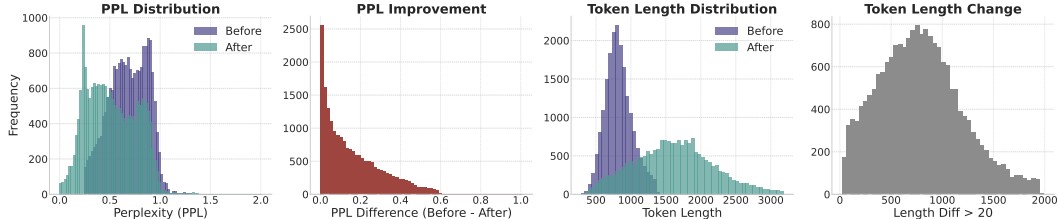

Figure 4: Analysis of Token Length & Perplexity Before and After the Search. The leftmost two plots show that our iterative search process consistently **reduces perplexity (PPL)**. The rightmost two plots show that the process also tends to **increase the token length** of the thinking trajectory, reflecting the addition of more detailed reasoning steps.

**Artistic** (Literature and Arts) writing, which is further broken down into sub-genres like Creative Writing and Essay Writing.

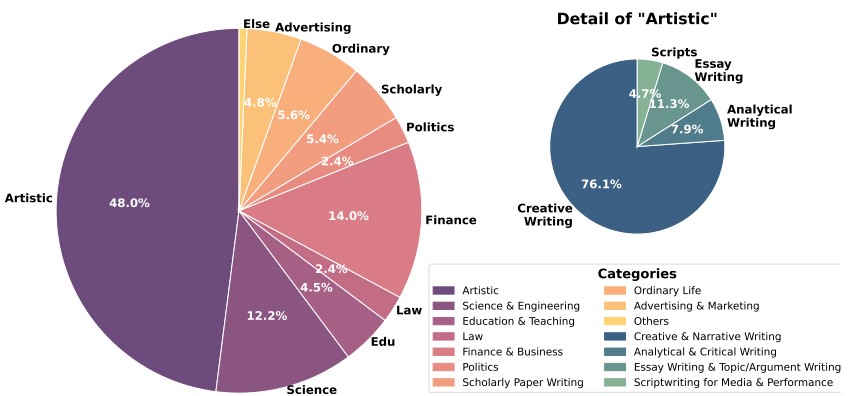

Figure 5: Distribution of the final 20K training dataset by categories taking more than 0.5% account. The primary chart shows a diverse range of topics, with a large emphasis on **Artistic** writing. The detailed view of "Artistic" reveals a focus on Creative Writing and other styles, ensuring comprehensive coverage for open-ended generation.

### 2.3.3 Final Dataset Assembly for Fine-Tuning.

Training a model exclusively on domain-specific data risks overfitting and can degrade its general knowledge priors. To mitigate this, we adopted a **mixed-data training strategy**. We combined our 20K synthetically generated trajectories with distilled deep reasoning trajectories from public datasets, i.e., OpenThoughts (Guha et al., 2025), that primarily cover domains like mathematics, coding, and science. This blended datasets prevents the model from catastrophic overfitting when learning deep reasoning for open-ended generation.

To train the base LLM, each complete triple in the final dataset was formatted using the prompt template shown in Listing 4 in the appendix. This structure explicitly teaches the model to first perform deep reasoning before producing the final output, thereby internalizing the desired reasoning process from scratch.

## 3 EXPERIMENTS

Our empirical evaluation is structured to rigorously validate the efficacy of DeepWriter. We address two central research questions:

1. How does DeepWriter, fine-tuned from scratch on an 8B open-source model, compare against state-of-the-art proprietary models and other powerful open-source alternatives across a spectrum of diverse open-ended generation tasks?

2. What is the individual contribution of each core component of our approach – specifically, the synthesized deep thinking trajectories, the iterative refinement algorithm, and the characteristics of the thinking traces and the data composition – to the model's final performance?

To answer these questions, we first present a comprehensive comparison against leading models, followed by a series of targeted ablation studies. We also provide a qualitative deep-dive analysis into the model's reasoning capabilities in the appendix.

**Training Data:** Our primary training dataset comprises approximately 20,000 deep thinking trajectories, which we synthesized for 16,000 unique queries spanning a wide array of open-ended tasks. As stated in Section 3, to prevent catastrophic forgetting of general reasoning abilities, we blended this core dataset with public thinking-process datasets that reasoning-related domains (e.g., mathematics, coding). This resulted in a final mixed dataset of 37,000 examples, ensuring a balance between specialized open-ended generation capabilities and keeping broad knowledge priors.

**Implementation Details:** We selected **Qwen3-8B-Base** as our base model for fine-tuning. This decision was informed by preliminary experiments where other candidates, such as Llama-3.1-8B-Base, struggled to effectively internalize the deep thinking process, and Qwen-2.5-7B-Base faced prohibitive context length limitations. For the critical trajectory synthesis stage, we utilized Qwen2.5-32B-Instruct as the generator model. Fine-tuning was conducted for 3 epochs using a constant learning rate of $2 \times 10^{-5}$ and a global batch size of 96. We set the max step to 10 and stopping PPL to 0.25 in the iterative local search procedure.

## 3.1 EVALUATION BENCHMARKS

To ensure a comprehensive and multi-faceted evaluation, we employed a suite of three complementary benchmarks: LongBench-Write (LB), HelloBench (HB), and WritingBench (WB). Together, they probe three distinct and critical dimensions of generative performance: raw endurance, real-world applicability, and domain-specific proficiency.

- **LongBench-Write (LB):** This benchmark functions as a targeted stress test for generative endurance. It is designed to measure a model's ability to produce coherent, ultra-long-form text (e.g., >10,000 words), allowing us to assess the foundational capacity for maintaining thematic consistency over extended outputs.
- **HelloBench (HB):** To gauge practical applicability, HelloBench evaluates performance on a diverse set of "in-the-wild" tasks sourced from real user queries. Our analysis focuses on two key subsets: **HB-A (Open-Ended QA)**, which tests the generation of detailed and nuanced answers, and **HB-B (Heuristic Text Generation)**, which assesses creative reasoning and stylistic fidelity in long-form narrative continuation.
- **WritingBench (WB):** This benchmark is tailored to measure domain-specific proficiency and controllability across six professional and creative domains: **A** (Academic & Engineering), **B** (Finance & Business), **C** (Politics & Law), **D** (Literature & Arts), **E** (Education), and **F** (Advertising & Marketing). It specifically evaluates the ability to adhere to complex, multi-dimensional constraints, a hallmark of advanced open-ended generation.

**Evaluation Protocols:** Given the subjective nature of open-ended tasks, we adopted the established protocol of using powerful LLMs as judges within each benchmarks[1]. While we acknowledge the potential for inherent biases in this method, it remains the most scalable and consistent approach for evaluating nuanced generative quality. Specifically, **Claude-3.7** was used to score outputs for LongBench and WritingBench, while **GPT-4o** was used for HelloBench. For HelloBench, we report the original score without rescaling.

## 3.2 MAIN RESULTS

We benchmarked DeepWriter against leading proprietary models (GPT-4o, Claude 3.5, Claude 3.7) and strong open-source baseline, Qwen2.5-32B-Instruct, Qwen3-8B, LongWriter-8B. The results, presented in Table 1, unequivocally demonstrate that our methodology successfully instills sophisticated generation capabilities in an 8B model without costly distillation or trial-and-error.

---

[1]Using the latest evaluation protocol of WritingBench, we note that there is currently discrepancy on reproduced results and paper results, which is also acknowledged by the authors.

Table 1: Main performance comparison on LongBench (LB), HelloBench (HB), and WritingBench (WB). DeepWriter demonstrates competitive performance against leading proprietary models and significantly outperforms other open-source models.

| Model | LB | HB-A | HB-B | WB-A | WB-B | WB-C | WB-D | WB-E | WB-F |
|---|---|---|---|---|---|---|---|---|---|
| GPT-4o | 83.1 | 83.7 | 87.6 | 74.4 | 73.4 | 74.3 | 77.9 | 75.8 | 78.0 |
| Claude 3.5 | 89.3 | 82.9 | 88.3 | 59.05 | 57.6 | 56.3 | 59.3 | 62.0 | 67.7 |
| Claude 3.7 | 97.8 | 83.9 | 93.2 | 78.2 | 77.9 | 76.5 | 79.3 | 79.2 | 80.8 |
| Qwen2.5-32B-Instruct | 78.8 | 81.0 | 83.8 | 52.5 | 49.8 | 51.0 | 49.6 | 53.9 | 54.2 |
| Qwen3-8B | 85.2 | 81.4 | 85.3 | 68.7 | 68.9 | 67.0 | 67.2 | 71.2 | 71.3 |
| LongWriter-8B | 76.5 | 80.1 | 82.6 | 57.9 | 53.9 | 49.0 | 52.0 | 52.9 | 52.0 |
| **DeepWriter-8B** | **91.3** | **82.6** | **87.4** | **72.2** | **71.8** | **69.8** | **70.6** | **73.7** | **72.3** |

Table 2: Ablation studies. The full model (top row) is compared against versions with key components removed. Results show that the synthesized deep thinking trajectories and iterative refinement are crucial for performance.

| Model Configuration | LB | HB-A | HB-B | WB-A | WB-B | WB-C | WB-D | WB-E | WB-F |
|---|---|---|---|---|---|---|---|---|---|
| **DeepWriter-8B (Full)** | **91.3** | **82.6** | **87.5** | **72.2** | **71.8** | **69.8** | **70.6** | **73.7** | **72.3** |
| - Remove Synthesis Data | 82.9 | 70.9 | 73.7 | 63.4 | 62.7 | 62.8 | 57.7 | 66.3 | 62.7 |
| - Remove Iterative Search | 83.2 | 81.0 | 84.4 | 66.7 | 68.7 | 67.3 | 65.6 | 69.5 | 70.1 |
| - Remove Reflection Tokens | 86.9 | 82.2 | 82.8 | 71.6 | 69.6 | 70.4 | 62.0 | 69.9 | 71.9 |
| - Downsample Long Traces | 90.3 | 82.2 | 84.0 | 69.6 | 70.3 | 69.1 | 67.5 | 69.8 | 70.7 |
| - Downsample Short Traces | 89.3 | 81.1 | 82.1 | 70.8 | 70.6 | 70.0 | 66.9 | 72.4 | 69.7 |
| - Remove Literature data | 88.8 | 81.6 | 85.3 | 71.3 | 71.0 | 69.3 | 69.8 | 72.2 | 71.3 |

**Analysis of Main Results.** The results in Table 1 reveal several compelling findings. First, **DeepWriter-8B consistently and substantially outperforms the strong open-source baseline, LongWriter-8B, across all benchmarks**. The performance gap is particularly stark in the diverse WritingBench domains, where DeepWriter achieves an average uplift of over 18 points. This highlights the profound advantage of our deep thinking synthesis approach over standard instruction tuning for cultivating advanced generative skills.

Second, **DeepWriter-8B closes a significant portion of the performance gap with elite proprietary models**. On the creative HelloBench task (HB-B), its score (87.48) is statistically on par with GPT-4o (87.6) and Claude 3.5 (88.3). More strikingly, on the professional writing tasks in WritingBench, DeepWriter-8B not only surpasses Claude 3.5 by a large margin in all six categories but also remains highly competitive with the much larger GPT-4o and Claude 3.7 models. A counter-intuitive result is DeepWriter-8B's score of 91.28 on LongBench-Write, exceeding both GPT-4o (83.1) and Claude 3.5 (89.3). This suggests that explicitly training on structured thinking trajectories provides a powerful inductive bias for maintaining long-range coherence, a critical challenge in ultra-long text generation.

## 3.3 ABLATION STUDIES

To meticulously dissect the contribution of each component of our methodology, we conducted a series of ablation studies, with results detailed in Table 2. Each experiment isolates a specific design choice to quantify its impact on overall performance.

The ablation results provide robust evidence supporting our methodological design.

- **Importance of Synthesized Data:** Removing our 20K synthesized trajectories and training only on public thinking datasets ("- Remove Synthesis Data") causes the most significant performance degradation across the board. Scores plummet, particularly in creative tasks like HelloBench HB-B ($87.48 \rightarrow 73.73$) and across WritingBench (average drop of over 8 points). This confirms a core hypothesis: it is not merely the presence of "thinking" data that matters, but the **quality and relevance of structured trajectories tailored for open-ended domains** that drive performance.

- **Impact of Iterative Refinement:** Using the initial, unrefined thinking trajectories ($z^{(0)}$) instead of the final, optimized ones ($z^*$) ("- Remove Iterative Search") also leads to a clear drop in performance. While the decline is less severe than removing the synthesis data entirely, the drop on nuanced WritingBench tasks (e.g., WB-A: $72.20 \rightarrow 66.72$) is substantial. This proves that our perplexity-guided local search is highly effective at discovering superior reasoning paths that translate directly into stronger generative capabilities.

- **Effect of Reflection Tokens:** Removing reflection tokens (e.g., 'Hmm...', 'Wait, that's...') from the synthesis prompts ("- Remove Reflection Tokens") has a nuanced effect. While overall scores dip slightly, the most pronounced drop is in WritingBench domain D (Literature & Arts), which falls from 70.57 to 62.04. This suggests that these explicit markers of cognitive exploration, self-correction, and branching are particularly valuable for instilling the flexibility and creativity required in artistic writing tasks.

- **Role of Trajectory Length:** We explored the impact of trace length by selectively downsampling either long or short trajectories. The results reveal a task-dependent preference: removing longer, more elaborate traces ("- Downsample Long Traces") disproportionately harms performance on complex, domain-specific tasks like those in WritingBench. Conversely, removing shorter, more concise traces ("- Downsample Short Traces") has a slightly larger negative impact on creative tasks like HB-B. This suggests that detailed, multi-step plans are crucial for structured professional writing, while nimbler, more direct reasoning may be optimal for creative ideation.

- **Role of Literature & Arts Data:** Removing the data from the "Literature & Arts" and "Ordinary Life" domains ("- without Literature & Arts data") degrades performance across all benchmarks, not just in the corresponding WB-D category. This finding indicates that training on creative and narrative tasks imparts a more generalizable ability to handle nuance, structure, and open-endedness, even benefiting performance in more technical domains. This highlights the contribution of the release of our 20K dataset covering comprehensive topics.

## 4 RELATED WORK AND FUTURE DIRECTIONS

The paradigm of "deep reasoning" aims to move beyond rapid, surface-level inference by leveraging increased computational investment at test time, a strategy shown to be effective by advanced models (Team et al., 2023; Guo et al., 2025; Jaech et al., 2024; Team, 2025; Muennighoff et al., 2025; Fu et al., 2025). This approach gained prominence with methods like Chain-of-Thought (CoT) prompting (Wei et al., 2022), which elicits intermediate reasoning steps, and has evolved into more sophisticated strategies like Tree-of-Thought (Yao et al., 2023) and self-refinement (Madaan et al., 2023; Kumar et al., 2024; Zelikman et al., 2022; 2024). While these techniques excel in verifiable domains like mathematics, their application to open-ended, creative tasks remains limited by the absence of a singular ground truth for verification. Our work, REverse-Engineered Reasoning (REER), directly addresses this gap.

Two dominant paradigms exist for instilling reasoning into a model's parameters: reinforcement learning (RL) and instruction distillation. RL is effective when clear reward signals are available (Ouyang et al., 2022; Guo et al., 2025; Team et al., 2025; Wang et al., 2025c), but struggles in creative domains where crafting a reward model to capture subjective qualities is an immense challenge in itself (Ouyang et al., 2022; Zhang et al., 2024; Lu, 2025). WritingZero (Lu, 2025) adopts this approach, but data and models remain closed. While recent work like VeriFree (Zhou et al., 2025) also uses a proxy for reward in verifiable domains, REER applies a similar principle to recover human-like reasoning for the broader challenge of open-ended generation.

Instruction Distillation offers an alternative, wherein reasoning traces are generated by a powerful but proprietary "teacher" model, e.g., GPT-4 (Achiam et al., 2023). While effective, this approach is often prohibitively expensive and is fundamentally capped by the capabilities of the "teacher" model (Guha et al., 2025; Toshniwal et al., 2024). To overcome these data bottlenecks, researchers have increasingly turned to synthetic data generation that builds a solution "forwards" (Wang et al., 2022; Zelikman et al., 2022; 2024), e.g., LongWriter (Wu et al., 2025). Our central innovation is to "reverse-engineer" reasoning by synthesizing it backwards from known good outcomes. By operationalizing this as a scalable, gradient-free search guided by perplexity, we created Deep Writing-20K, the first large-scale dataset of deep reasoning trajectories for open-ended tasks.

Our model, Deep Writer-8B, trained on this data, validates REER as a powerful and cost-effective method, surpassing strong open-source baselines and achieving performance competitive with leading proprietary models like GPT-4o. Importantly, our creation and release of Deep Writing-20K also democratizes access to high-quality deep reasoning data, addressing a critical bottleneck for the research community. **This opens several promising directions for future research.** A primary avenue is scaling our experiments to larger models and datasets to investigate the scaling laws of reverse-engineered reasoning. Furthermore, the core principle of REER is well-suited for novel scenarios where reasoning annotations are scarce, making it a valuable paradigm to explore in complex domains such as multi-step agentic tasks, scientific discovery, and multi-modal reasoning.

## ETHICS STATEMENT

This research adheres to the ICLR Code of Ethics. Our work centers on the development of a new methodology for training large language models and the creation of a new dataset, 'Deep Writing-20K'. The data for this dataset was sourced from publicly available and permissible sources, including online communities (e.g., r/WritingPrompts), public domain texts (Project Gutenberg), and existing open-source datasets (WildChat, LongWriter6K). We have strictly filtered on top of theses datasets to ensure that our data collection and usage practices respect user privacy and do not include personally identifiable information.

The primary goal of this research is to advance the understanding of reasoning in AI for open-ended, creative, and professional tasks. However, we acknowledge that, like any powerful generative model, the methods and models presented could be misused for generating harmful, biased, or misleading content. We tried our best to filter out harmful contents and prevented the model from internalizing implicit societal biases.

The models and datasets used in our research, including the Qwen model series, are used in accordance with their respective licenses. We intend for our open-sourced dataset, 'Deep Writing-20K', to be used by the research community to foster further investigation into transparent and beneficial AI reasoning mechanisms.

## REPRODUCIBILITY STATEMENT

We are committed to ensuring the reproducibility of our research. All components required to replicate our findings are detailed within the paper and its appendix, and we provide a preview of the data in the supplementary.

The detailed data curation pipeline, including sourcing, synthesis, and filtering procedures, is described in Section 2.3. The core algorithm for 'Reverse-Engineered Reasoning (REER)' via iterative local search is detailed in Section 2.2. The exact prompts used for generating initial trajectories, performing segment-wise edits, and conducting inference are provided in the Appendix, Listings 1-4. The code is attached in the supplementary.

We used publicly available base models for our experiments. The trajectory synthesis was performed using 'Qwen2.5-32B-Instruct', and the final 'Deep Writer-8B' model was fine-tuned from 'Qwen3-8B-Base'. All training hyperparameters, including learning rate, batch size, and number of epochs, are specified in Section 3.1 .

Our evaluations were conducted on publicly available benchmarks: LongBench-Write, HelloBench, and WritingBench. The evaluation protocols, including the LLMs used as judges, are described in Section 3.2.

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

## A  LIST OF PROMPTS

Below we list the exact prompts used for trajectory synthesis and in-house evaluation. For the meta-structure guidelines and thinking pattern injection, refer to Listing 1. For enforcing segment-wise edits, refer to Listing 2. For quality assessment with regard to deep reasoning, refer to Listing 4. For computing the proxy score of a deep reasoning trajectory, we employ Listing 3, without including the reference output.

## LISTINGS

Listing 1: Prompt for Generating Initial Thinking.

```
1  You are an expert in many fields. Suppose you will give a specific
       final response, I need you to also write down the thought
      process behind this solution.
2  Here is a task:
3  {}
4
5  Here is the solution you will create:
6  {}
7
8  Now, you need to write down the thinking process behind this
       solution, as if you are thinking aloud and brainstorming in
       the mind. The thinking process involves thoroughly exploring
       questions through a systematic long thinking process. This
       requires engaging in a comprehensive cycle of analysis,
       summarizing, exploration, reassessment, reflection,
       backtracing, and iteration to develop well-considered thinking
        process. Present your complete thought process within a
       single and unique `<think></think>` tag.
9
10 Your thought process must adhere to the following requirements:
11
12 1.  **Narrate in the first-person as if you are thinking aloud and
        brainstorming**
13     Stick to the narrative of "I". Imagine you are brainstorming
      and thinking in the mind. Use verbalized, simple language.
14
15 2.  **Unify the thinking process and the writing solution:**
16     Your thought process must precisely correspond to a part of
      the writing solution. The reader should be able to clearly see
       how your thoughts progressively "grew" into the finished
      piece, making the copy feel like the inevitable product of
      your thinking.
17
18 3.  **Tone of Voice: Planning, Sincere, Natural, and Accessible**
19     Imagine you are analyzing and planning what to do before you
      start to wrtie the solution.  Your language should be plain
      and easy to understand, avoiding obscure professional jargon
      to explain complex thought processes clearly.
20
21 4.  **Logical Flow: Clear and Progressive**
```

```
22
23   5.  **Thinking Framework for deep thinking**
24        To ensure your thinking is clear and deep, to showcase your
           thinking and planning to fulfill the task, below is what you
           might cover when you are thinking aloud and brainstorming.
25
26        Understanding the user intent and the task: Before putting pen
           to paper, I need to thoroughly consider the fundamental
           purpose of the writing. I first need to discern the user's
           true goal behind their literal request. Next, I will consider:
           Who am I talking to? I will create a precise profile of the
           target reader, understanding their pain points, aspirations,
           and reading context. Then, I will establish the Core Objective:
           What specific emotional, cognitive, and behavioral changes do
           I most want the reader to experience after reading?
27
28        Establishing the content: I need to brainstorm a core creative
           idea and communication strategy centered around my objective.
           Then, I will think about what content and key information I
           need to convey to the reader to fulfill the writing task, and
           what source materials this will involve.
29
30        Building the structure: I need to design a clear narrative
           path for the reader, like a "blueprint." First, I will plan
           the article's skeleton (e.g., using a framework like the
           Golden Circle "Why-How-What," the AIDA model "Attention-
           Interest-Desire-Action," or a narrative structure "Beginning-
           Development-Climax-Resolution"). Then, I will plan the key
           modules: How will the introduction hook the reader? How will
           the body be layered and the arguments arranged? How will the
           conclusion summarize, elevate the message, and provide a clear
           Call to Action (CTA)?
31
32        Outline: If the task output might be relatively long, I will
           consider writing an outline (or a draft) which naturally
           derives from the plan above. Specifically, the outline will
           ground my plan into paragraphs, summarizing the key content
           for each paragraph and what are the key points here, sentence
           structure or anything important for the paragraph.
33        I PROMISE I will NOT copy the solution I will NOT copy the
           solution, this outline (or draft) should only look like a
           prototype or outline of the target text. After finishing this
           outline, I will check again if there are any details or notes
           I should pay attention to when writing the final solution.
34        I will begin writing this draft after a `--- Outline (or Draft
           ) ---` separator at the end of my thinking process. The draft
           will be included in the same `<think></think>` block.
35
36
37   6. Throughout the thinking process, I want to involve deep
          thinking and planning, and use deliberate self-critique/self-
          reflection in my thinking process. Trigger these by regularly
          using patterns such as `wait`, `maybe`, `let me`, etc. For
          example:
38        - Hmm, maybe .. (other concrete thinking regarding the given
          request)
39        - Let me think ..
40        - Wait no ..
```

```
41        - But wait ..(might find something wrong with your previous
      thoughts)
42        - Wait, that's a bit ..(reflections about previous decisions).
       Let me think .. (are thinking of other possibilities)
43        - Wait, the user said ..(backtracing of previous information).
       So ..
44        - Hmm...Alternatively, maybe ..(branching on other
      possibilities)
45        - But ..
46   But I promise I will use diverse triggers and will NOT use same
      triggers repeatedly. I will use these when analyzing user
      needs, establishing content and structure and when I consider
      alternatives, backtracing and the details. I will NOT use them
       when I write the draft or I am approaching the end of
      thinking.
47
48   In the thinking process, make sure NO PAST TENSES, NO PAST TENSES,
       because this is the thought process before you are to write a
       final solution. You are planning what you will and you need
      to do.
49   Imagine you're thinking aloud and brainstorming. Write it as an
      internal monologue or a stream of consciousness. Do not use
      bullet points, numbers, or formal section headings.
50   Now record your thinking process within '<think></think>' tags.
```

Listing 2: Prompt for Segment-wise Edits.

```
1    Your task is to receive a user request, a target output, and an
      existing thinking process, and then to refine and enrich a
      specific paragraph within that thinking process.
2
3    ---> **Task**
4    {}
5    ---> **Target Output**
6    {}
7    ---> **Thinking Process**
8    {}
9    <replace>
10   {}
11   </replace>
12   {}
13
14   Follow this three-step method to construct your response:
15
16   **Step 1:** Locate the paragraph you need to revise within the
      existing thinking process. In relation to the surrounding
      context, what is the primary function of this paragraph?
17
18   **Step 2:** Read the "Target Text" and the "paragraph to be
      revised" side-by-side. Ask yourself a key question: Does the
      thinking process reflected in this paragraph lack crucial
      steps, or is there content that could be further optimized and
       detailed to better align with the Target Text?
19
20   **Step 3:** Improve and optimize the paragraph (that represents a
      part of the thinking process).
21   - Based on the analysis, refine the initial target paragraph into
      a new one, base remain the tone. Put the refinement into <
      refine></refine> tags.
```

```
22  - To help involve deep thinking and planning, consider deliberate
       self-critique/self-reflection in your thinking process.
       Trigger these by frequently using patterns such as `wait`, `
       maybe`, `let me`, etc. For example:
23     - Hmm, maybe .. (other concrete thinking regarding the given
       request)
24     - Let me think ..
25     - Wait no ..
26     - But wait ..(might find something wrong with your previous
       thoughts)
27     - Wait, that's a bit ..(reflections about previous decisions).
        Let me think .. (are thinking of other possibilities)
28     - Wait, the user said ..(backtracking of previous information)
       . So ..
29     - Hmm...Alternatively, maybe ..(branching on other
       possibilities)
30     - But ..
31  - If the function of the paragraph being improved is to serve as a
        first draft of the text, you must focus on enhancing the text'
       s logic and completeness. The draft should not be a general
       outline but should express specific content and state a clear
       point of view. Consider whether the current draft is an
       appropriate prototype for the Target Text: it should be
       neither too vague nor a direct copy, but should reflect a
       foundational version.
32
33  Based on the guide above, you are to refine **only** the section
       marked for replacement below.
34  <replace>
35  {}
36  </replace>
37
38  In your response, first, present your analysis following the three
       -step method within `<analyze></analyze>` tags. Finally, place
        the corresponding, refined paragraph of the **thinking
       process** within `<refine></refine>` tags.
39  Notes: a. Avoid repeating. Reduce the use of the same connection
       words, avoid repeating the same meanings over and over again.
       Ensure that your revised content does not repeat information
       from the context.
40  b. please keep the first a few words of the original paragraph,
       especially the connection words
41  c. use self-critique trigger words, such as `wait`, `maybe`, `let
       me`, etc.
```

Listing 3: Prompt for Standard Inference.

```
1
2   You are an expert in many fields. Suppose you will give a specific
        final response, I need you to also write down the thought
       process behind this solution.
3   Here is a task:
4   {}
5
6   Now, you need to think aloud and brainstorm in the mind. The
       thinking process involves thoroughly exploring questions
       through a systematic long thinking process. This requires
       engaging in a comprehensive cycle of analysis, summarizing,
       exploration, reassessment, reflection, backtracing, and
```

```
      iteration to develop well-considered thinking process. Present
       your complete thought process within a single and unique '<
      think></think>' tag.
  7
  8  Your thought process must adhere to the following requirements:
  9
 10  1.  **Narrate in the first-person as if you are thinking aloud and
         brainstorming**
 11     Stick to the narrative of "I". Imagine you are brainstorming
        and thinking in the mind. Use verbalized, simple language.
 12
 13  2.  **Unify the thinking process and the writing solution:**
 14     Your thought process must precisely correspond to a part of
        the writing solution. The reader should be able to clearly see
         how your thoughts progressively "grew" into the finished
        piece, making the copy feel like the inevitable product of
        your thinking.
 15
 16  3.  **Tone of Voice: Planning, Sincere, Natural, and Accessible**
 17     Imagine you are analyzing and planning what to do before you
        start to wrtie the solution.  Your language should be plain
        and easy to understand, avoiding obscure professional jargon
        to explain complex thought processes clearly.
 18
 19  4.  **Logical Flow: Clear and Progressive**
 20
 21  5.  **Thinking Framework for deep thinking**
 22     To ensure your thinking is clear and deep, to showcase your
        thinking and planning to fulfill the task, below is what you
        might cover when you are thinking aloud and brainstorming.
 23
 24      Understanding the user intent and the task: Before putting pen
         to paper, I need to thoroughly consider the fundamental
        purpose of the writing. I first need to discern the user's
        true goal behind their literal request. Next, I will consider:
         Who am I talking to? I will create a precise profile of the
        target reader, understanding their pain points, aspirations,
        and reading context. Then, I will establish the Core Objective:
         What specific emotional, cognitive, and behavioral changes do
         I most want the reader to experience after reading?
 25
 26      Establishing the content: I need to brainstorm a core creative
         idea and communication strategy centered around my objective.
         Then, I will think about what content and key information I
        need to convey to the reader to fulfill the writing task, and
        what source materials this will involve.
 27
 28      Building the structure: I need to design a clear narrative
        path for the reader, like a "blueprint." First, I will plan
        the article's skeleton (e.g., using a framework like the
        Golden Circle "Why-How-What," the AIDA model "Attention-
        Interest-Desire-Action," or a narrative structure "Beginning-
        Development-Climax-Resolution"). Then, I will plan the key
        modules: How will the introduction hook the reader? How will
        the body be layered and the arguments arranged? How will the
        conclusion summarize, elevate the message, and provide a clear
         Call to Action (CTA)?
 29
```

```
30      Draft: unless it is a really easy request, otherwise I need to
         consider writing a draft based on the plan above, before you
         give the final writing solution.  I will translate my plan
         into paragraphs, considering the key points, content, and
         sentence structure for each. This initial draft should look
         like a prototype of the target text. This draft will be way
         shorter than the final writing solution within controlled
         length, but it must also avoid being too vague or general or
         simply copying the final text. I will begin writing this draft
          after a `--- The Draft ---` separator at the end of my
         thinking process. The draft will be included in the same `<
         think></think>` block. After writing the draft, I will further
          critique what can be improved, and analyze what details can
         be enriched (and hence make it more likely to eventually
         arrive at the given solution)
31
32   6. Throughout the thinking process, I want to involve deep
         thinking and planning, and use deliberate self-critique/self-
         reflection in my thinking process. Trigger these by frequently
          using patterns such as `wait`, `maybe`, `let me`, etc. For
         example:
33      - Hmm, maybe .. (other concrete thinking regarding the given
         request)
34      - Let me think ..
35      - Wait no ..
36      - But wait ..(might find something wrong with your previous
         thoughts)
37      - Wait, that's a bit ..(reflections about previous decisions).
          Let me think .. (are thinking of other possibilities)
38      - Wait, the user said ..(backtracking of previous information)
         . So ..
39      - Hmm...Alternatively, maybe ..(branching on other
         possibilities)
40      - But ..
41
42   Now record your clear, complete, and logical thinking process
         within `<think></think>` tags.
43   In the thinking process, make sure NO PAST TENSES, NO PAST TENSES,
          because this is the thought process before you are to write a
          final solution. You are planning what you will and you need
         to do.
44   Imagine you're thinking aloud and brainstorming. Write it as an
         internal monologue or a stream of consciousness. Do not use
         bullet points, numbers, or formal section headings.
```

Listing 4: Prompt for Rating Response Quality w.r.t. Deep Reasoning.

```
1
2
3   You are an expert judge in AI generated content. Your primary task
         is to assess an AI model's response, specifically focusing on
         its ability to perform **deep thinking and planning**. You
        will evaluate the response across five distinct dimensions. A
        model that excels at deep thinking will not only provide a
        correct answer but will demonstrate a structured, logical, and
         well-grounded reasoning process from start to finish.
4
5   Your final output must be a structured report with a score and
         justification for each dimension.
```

```
6
7   -----
8
9   ## Evaluation Dimensions & Scoring
10
11  ### 1\. Understanding & Problem Decomposition
12
13  **Relation to Deep Thinking:** This is the foundational step. Deep
        thinking is impossible without first accurately understanding
        the problem in its entirety. This dimension measures if the
        model comprehends the user's explicit and implicit needs and
        then breaks down the complex request into manageable, logical
        parts. This act of decomposition *is* the first stage of
        planning.
14
15    * **Score 1 (Poor):** The model fundamentally misunderstands the
         user's request or ignores key components. The response is off-
        topic or fails to address the core problem.
16    * **Score 3 (Average):** The model grasps the main goal but may
        overlook nuances or implicit constraints. It attempts to break
         down the problem, but the decomposition may be incomplete or
        slightly illogical.
17    * **Score 5 (Excellent):** The model demonstrates a
        comprehensive understanding of the user's intent, including
        subtle details. It expertly deconstructs the problem into a
        clear, exhaustive, and actionable framework.
18    Score 2 and Score 4 fit interpolate into the above scoring
        criterion.
19  -----
20
21  ### 2\. Content Structure & Logical Consistency
22
23  **Relation to Deep Thinking:** This dimension reflects the clarity
         and order of the model's thought process. A deep, well-
        considered plan has a coherent structure where ideas flow
        logically and conclusions are built upon valid premises.
        Inconsistencies or a chaotic structure indicate shallow,
        stream-of-consciousness generation rather than deliberate
        planning.
24
25    * **Score 1 (Poor):** The response is disorganized, rambling, or
         internally contradictory. It's difficult to follow the model'
        s line of reasoning.
26    * **Score 3 (Average):** The response has a discernible
        structure (e.g., uses headings, lists), but the flow between
        sections could be improved. It is mostly consistent, with only
         minor logical gaps.
27    * **Score 5 (Excellent):** The response is impeccably structured
        . Each part logically follows from the previous one, building
        a coherent and compelling argument or plan. The internal logic
         is sound and easy to follow from beginning to end.
28    Score 2 and Score 4 interpolate into the above scoring criterion
        .
29  -----
30
31  ### 3\. Depth of Analysis & Synthesis
32
33  **Relation to Deep Thinking:** This is the core of "deep thinking
        ." It goes beyond simply retrieving facts and measures the
```

```
              model's ability to analyze underlying principles, connect
              disparate ideas, and synthesize them to create new insights. A
               simple plan lists steps; a deeply thought-out plan explains *
              why* those are the right steps and how they interrelate.
  34
  35      * **Score 1 (Poor):** The response is superficial, relying on
              cliches or surface-level information. It shows no evidence of
              analyzing the "why" behind the "what."
  36      * **Score 3 (Average):** The model provides a competent analysis
              , explaining concepts correctly but treating them in isolation.
               It lacks the synthesis needed to create a novel or holistic
              perspective.
  37      * **Score 5 (Excellent):** The model provides a profound
              analysis, connecting concepts in insightful ways. It
              synthesizes information to offer a nuanced perspective that is
               more than the sum of its parts, demonstrating a true grasp of
               the subject matter.
  38      Score 2 and Score 4 interpolate into the above scoring criterion
              .
  39  -----
  40
  41  ### 4\. Presentation Clarity
  42
  43  **Relation to Deep Thinking:** A brilliant plan is useless if it
          cannot be understood. This dimension assesses the model's
          ability to communicate its complex thoughts and plans
          effectively. Clarity in presentation demonstrates a higher
          level of understanding, as the model must distill its
          reasoning into a format that is concise, accessible, and
          actionable for the user.
  44
  45      * **Score 1 (Poor):** The response is convoluted, filled with
              jargon, or poorly formatted. The user would struggle to
              understand the main points or how to act on the advice.
  46      * **Score 3 (Average):** The response is generally
              understandable but could be more concise or better organized.
              It may be overly dense or require the user to re-read sections
               to grasp the meaning.
  47      * **Score 5 (Excellent):** The response is exceptionally clear,
              concise, and well-formatted. It uses plain language and
              effective formatting (like lists, bolding, or tables) to make
              complex information easy to digest and act upon.
  48      Score 2 and Score 4 interpolate into the above scoring criterion
              .
  49  -----
  50
  51  ### 5\. Factual Grounding (Hallucination Check)
  52
  53  **Relation to Deep Thinking:** Deep thinking and planning must be
          grounded in reality to be useful. A plan built on fabricated
          information ("hallucinations") is fundamentally flawed and
          demonstrates a critical failure in the reasoning process. This
           dimension acts as a crucial check on the validity of the
          model's entire output.
  54
  55  *This dimension is scored on a severity scale, not a quality scale
          .*
  56
```

```
57     * **Score 4 (Factually Sound):** The response contains no
        discernible factual errors or hallucinations.
58     * **Score 3 (Minor Inaccuracy):** Contains a small error (e.g.,
        a slightly incorrect date, a minor misstatement) that does not
         undermine the overall logic or conclusion of the response.
59     * **Score 2 (Significant Hallucination):** Contains a major
        factual error that invalidates a key part of the argument or
        plan. The response is partially unreliable.
60     * **Score 0 (Critical Hallucination):** The core premise or a
        critical component of the response is based on a fabrication,
        rendering the entire output untrustworthy and potentially
        harmful.
61     Score 1 interpolates into the above scoring criterion.
62   -----
63
64   ## Final Output Format
65
66   Please provide your evaluation in the following structured json
        format.
67   ```json
68   {
69     "evaluationReport": {
70       "understandingAndDecomposition": {
71         "score": "[Enter a score from 1-5]",
72         "justification": "[Your justification here. Explain why you
        gave this score.]"
73       },
74       "structureAndConsistency": {
75         "score": "[Enter a score from 1-5]",
76         "justification": "[Your justification here. Explain why you
        gave this score.]"
77       },
78       "depthOfAnalysis": {
79         "score": "[Enter a score from 1-5]",
80         "justification": "[Your justification here. Explain why you
        gave this score.]"
81       },
82       "presentationClarity": {
83         "score": "[Enter a score from 1-5]",
84         "justification": "[Your justification here. Explain why you
        gave this score.]"
85       },
86       "factualGrounding": {
87         "severityScore": "[Enter a severity score from 1-5]",
88         "justification": "[Describe any inaccuracies or
        hallucinations found. If none, state 'Response is factually
        sound.']"
89       },
90       "overallSummary": "[Provide a final, concise paragraph
        summarizing the model's overall performance in deep thinking
        and planning. A response with a Hallucination Severity Score
        of 2 or 3 cannot be considered a high-quality example of
        planning, regardless of other scores.]"
91     }
92   }
93
94   ----
95   <User Request>
96
```

```
97   $INST$
98
99   </User Request>
100
101  <Response>
102
103  $RESPONSE$
104
105  </Response>
106  ----
107  Now go back to the evaluation guideline and give the json report
       ."""
```

## B    QUALITATIVE ANALYSIS

### B.1    GENERATION QUALITY OF DEEP THINKING

Beyond quantitative scores, we sought to understand how well DeepWriter internalizes the *qualities* of deep thinking. To this end, we conducted a qualitative analysis, scoring model outputs on five dimensions intrinsically linked to advanced reasoning and planning:

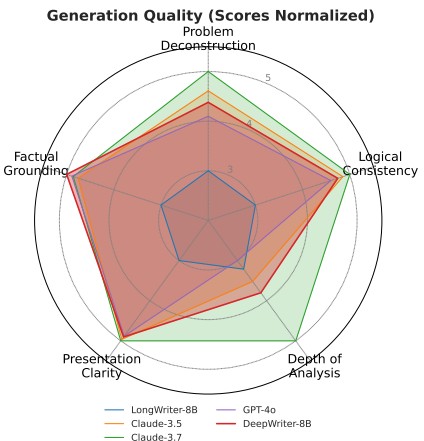

- **Problem Deconstruction:** The ability to break down a complex prompt into a logical hierarchy of sub-goals. This is the foundation of effective planning.

- **Logical Consistency:** Maintaining a coherent and non-contradictory reasoning path throughout the entire generation necessistates the ability to plan over the generation.

- **Depth of Analysis:** Moving beyond surface-level responses to explore nuances, consider alternatives, and demonstrate sophisticated understanding. This reflects the "deep" aspect of the thinking process.

- **Presentation Clarity:** The ability to structure the final output in a clear, organized, and persuasive manner, which is a direct outcome of a well-formed internal plan.

- **Factual Grounding:** Ensuring that generated content, where applicable, is accurate and well-supported, reflecting a robust and reality-aware reasoning process.

Figure 6: Qualitative comparison of generation quality. Scores are normalized across five dimensions related to deep thinking. DeepWriter-8B shows a reasoning profile far superior to the open-source baseline and is competitive with top proprietary models.

The normalized scores, visualized in the radar chart in Figure 6, provide a signature of each model's reasoning profile. As illustrated in Figure 6, DeepWriter-8B exhibits a remarkably strong and well-rounded reasoning profile. Its performance polygon significantly envelops that of the LongWriter-8B baseline, showing dramatic improvements across all five dimensions. This confirms that our methodology genuinely enhances underlying reasoning capabilities, rather than just improving superficial output fluency.

Furthermore, DeepWriter-8B's profile closely rivals that of GPT-4o and substantially exceeds Claude 3.5, particularly in **Depth of Analysis** and **Factual Grounding**. While the state-of-the-art Claude 3.7 still defines the frontier, especially in Depth of Analysis, our 8B model has demonstrably bridged a large portion of the capability gap. This validates our central claim: instilling a deep thinking process through gradient-free synthesis is a highly promising pathway toward building more powerful and scalable models.

## B.2 QUALITATIVE COMPARISON OF THINKING PATTERNS

To better understand how injecting human-like thinking patterns during synthesis affects the model's behavior, we analyze the frequency of reasoning phrases generated by the full model versus the ablated model trained without injecting thinking patterns. The thinking patterns are deduplicated such that occurances will be counted only once for the same solution.

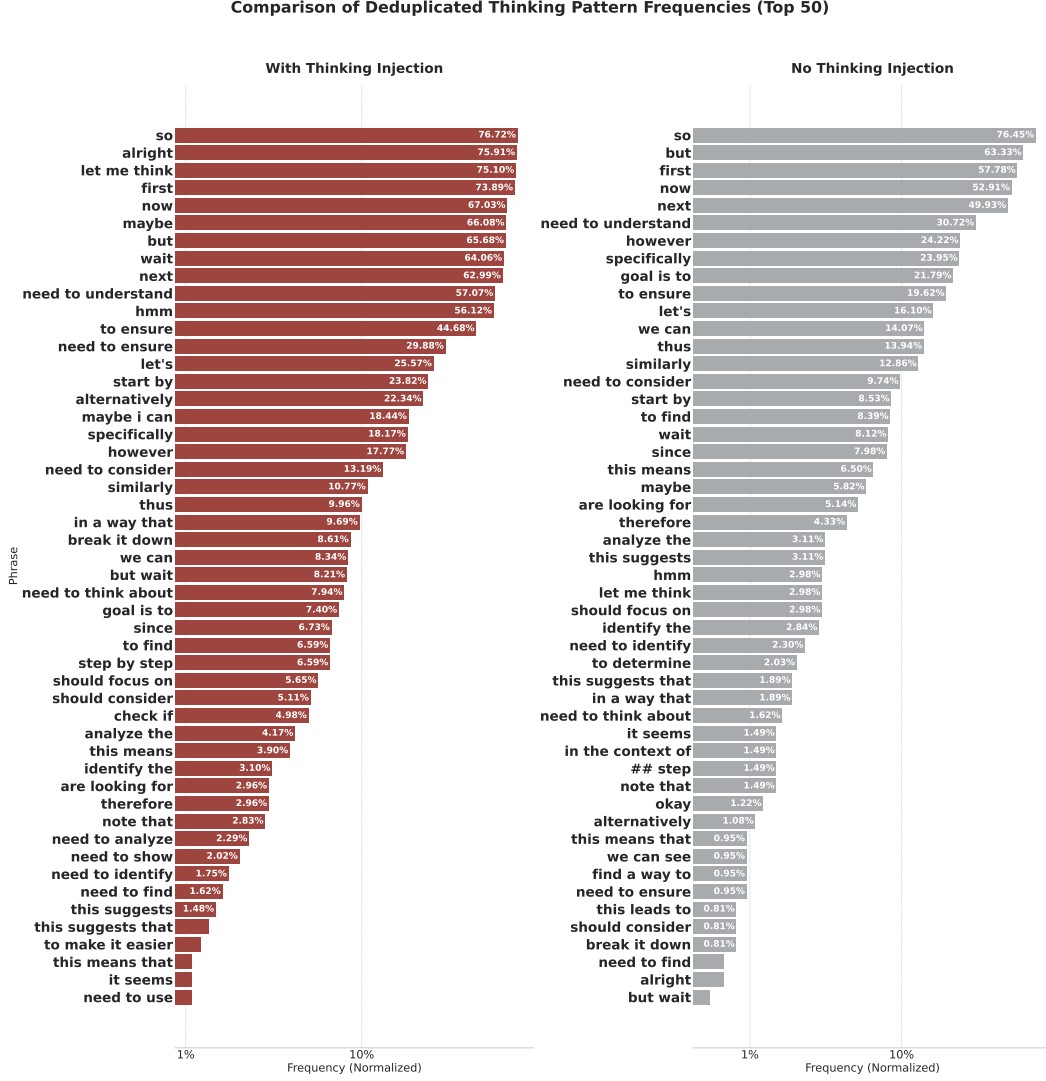

Figure 7: Comparison of the top 50 thinking pattern frequencies for models trained with and without the injection of human-like thinking patterns during data synthesis. The model with injection (left) shows a more diverse and balanced distribution of patterns, while the model without (right) relies heavily on a few formulaic phrases.

As shown in Figure 7, the difference is stark. The model trained *with* thinking pattern injection exhibits a more diverse and evenly distributed use of thinking patterns. Tokens indicating reflection and self-correction, such as 'let me think', 'maybe', 'hmm', and 'wait', are prominent. This suggests a more flexible, human-like reasoning process with cognitive exploration. In contrast, the model trained *without* this injection relies on a small set of highly frequent phrases like 'next', 'first', and 'goal is to'. The frequency distribution is highly skewed, indicating a more rigid and formulaic reasoning process.

This analysis confirms that the proposed context engineering techniques encourages the model to adopt a more nuanced and reflective approach to problem-solving, which, as shown in the ablation studies, is particularly beneficial for creative and complex tasks.

## C  BEHAVIORAL ANALYSIS

We conducted preliminary analysis on the model's behaviors. Figure 8 shows the token length distribution of DeepWriter-8B responses on LongBench-Write.

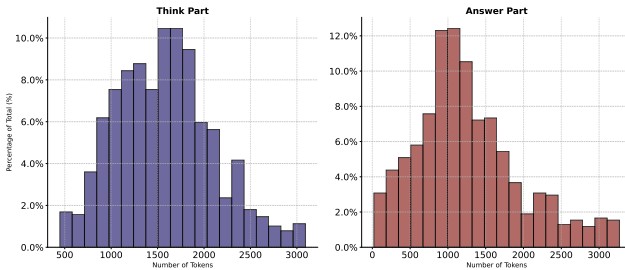

Figure 8: Token Length distribution of Thinking and Answer part of DeepWriter-8B.

We also compare the response string length distribution across leading models in Figure 9. While DeepWriter achieves superior performance competitive with frontier models, it does not introduce excessive response length like LongWriter. The average response length is around 5000 tokens, comparable with frontier models like GPT-4o and Claude-3.7.

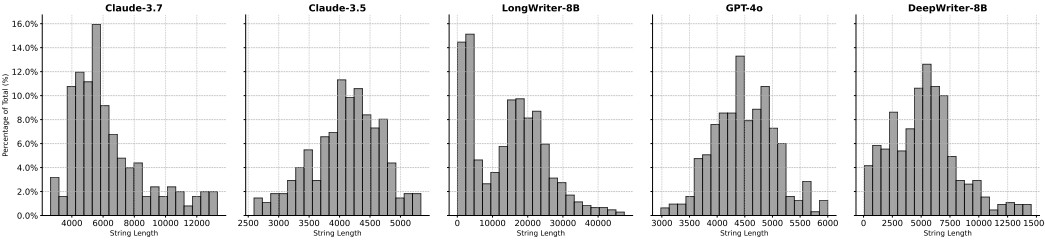

Figure 9: Response String Length Distribution across different models.

## D  CASE STUDIES

Due to formatting issues with latex, we put a few case studies in the supplementary materials. We manually review the cases where DeepWriter can outperforms other models, confirming the argument that DeepWriter can achieve better depth, logical consistency and factual grounding via deep reasoning traces. We also analyze the error patterns of the generations from DeepWriter. We find that DeepWriter often gets lower score due to domain knowledge gap, implying the benefits of training with more diverse corpus of topics and domains.

## E  EXTENDED RELATED WORK

**Deep Reasoning and Test-Time Computation.** The paradigm of "deep reasoning" (or Long CoTs) aims to move beyond rapid, surface-level inference by leveraging increased computational investment at test time. Advanced models from organizations like Google (Team et al., 2023), DeepSeek AI (Guo et al., 2025), and OpenAI (Jaech et al., 2024) have demonstrated the effectiveness of this test-time scaling (Team, 2025; Muennighoff et al., 2025; Fu et al., 2025). This approach gained prominence with methods like Chain-of-Thought (CoT) prompting (Wei et al., 2022), which elicits intermediate reasoning steps to guide a model toward more accurate solutions. Building on this,

more sophisticated strategies have emerged, such as Tree-of-Thought (ToT) (Yao et al., 2023), which explores a tree of possible reasoning paths, and various self-correction or self-refinement (Madaan et al., 2023; Kumar et al., 2024; Zelikman et al., 2022; 2024) mechanisms that iteratively improve an initial response. While these approaches have yielded remarkable performance gains in verifiable domains like mathematics and programming, their application to open-ended, creative tasks remains largely unexplored due to the absence of a singular ground truth for verification. REER addresses this gap by developing a method to instill this deliberate, structured thinking capability for non-verifiable creative domains.

**Paradigms for Instilling Reasoning.** Beyond prompting techniques at inference time, two dominant paradigms exist for integrating advanced reasoning capabilities directly into a model's parameters: reinforcement learning and instruction distillation.

Reinforcement Learning (RL) has been instrumental in aligning LLMs with human preferences (RLHF) and improving performance on tasks with clear reward signals (Ouyang et al., 2022; Guo et al., 2025; Team et al., 2025; Wang et al., 2025c). In verifiable domains, a correct outcome provides a straightforward positive reward, effectively guiding the model's search through a vast solution space (Shao et al., 2024; Wang et al., 2025a;b; Su et al., 2025). However, this reliance on verifiability presents a formidable barrier when applied to open-ended generation (Ouyang et al., 2022; Lu, 2025). Crafting a reward model that can reliably approximate nuanced and subjective qualities like originality or emotional resonance is an immense challenge in itself (Ouyang et al., 2022; Zhang et al., 2024). Furthermore, the subsequent RL process is often computationally burdensome and sample-inefficient (Shao et al., 2024; Gulcehre et al., 2023; Wang et al., 2023). Recently VeriFree (Zhou et al., 2025) extends verification-based reward to likelihood-based reward for reinforcement learning on verifiable domains. Likewise, REverse-Engineered Reasoning (REER) shares the principle of using a proxy to judge the reasoning quality. However, the motivation is fundamentally different – we focus on recovering human-like deep reasoning from known-good outputs for the broader open-ended generation problems.

Instruction Distillation offers an alternative, wherein reasoning traces are generated by a powerful "teacher" model (e.g., GPT-4 (Achiam et al., 2023)) and used as training data for a smaller "student" model. While effective, this approach is constrained by two fundamental limitations. First, it is often hampered by the prohibitive cost of querying state-of-the-art proprietary models at scale (Guha et al., 2025; Toshniwal et al., 2024). Second, and more fundamentally, distillation is capped by the teacher's abilities—a student model cannot learn a capacity that the teacher does not already possess (Toshniwal et al., 2024). This limitation is exacerbated by the general scarcity of high-quality, open-source instruction data tailored for advanced creative tasks (Bai et al., 2024).

To overcome these data bottlenecks, researchers have increasingly turned to synthetic data generation. Most approaches use a powerful LLM to generate new query-response pairs, often to augment existing datasets or bootstrap capabilities in new domains (Wang et al., 2022; Zelikman et al., 2022; 2024; Gu et al., 2025; Yang et al., 2023; Han et al., 2025). These methods aim to build a solution "forwards" for a given query through data synthesis. Our central innovation is to "reverse-engineer" reasoning – synthesize deep reasoning "backwards" from a known good outcome such as human-written solutions.

**Writing Datasets, Models and Benchmarks** Prior work has explored both synthetic data pipelines and RL in AI writing. For instance, Weaver (Wang et al., 2024) proposed instruction back-translation, LongWriter (Bai et al., 2024) proposed an agentic data pipeline to synthesize long-form writing outputs and introduced the LongBench-Write benchmark. In contrast, Writing-Zero (Lu, 2025) employed an RL approach, training a reward model on private datasets, but its training data remains unreleased. DeepWriter, to our knowledge, is the first to *instill deep reasoning for open-ended generation* using a scalable, open synthetic data approach.

Evaluation in this domain relies on recently developed benchmarks. HelloBench (Que et al., 2024) proposes a diverse collection of "in-the-wild" tasks from real user queries to gauge practical applicability. Meanwhile, WritingBench (Wu et al., 2025) measures domain-specific proficiency and the ability to adhere to complex, multi-dimensional constraints across six professional domains.

## F    LLM USAGE

We acknowledge the use of large language models (LLMs) during the preparation of this manuscript. The application of these tools was strictly limited to an assistive role for improving the quality of the writing. Specifically, LLMs were utilized to enhance clarity, refine sentence structure, and ensure a smooth and logical flow of arguments throughout the paper. The core ideas, methodology, experimental results, and all intellectual contributions presented herein are entirely the work of the authors. LLMs were not used to generate any of the substantive research content or analysis.

