# OpenReview forum: "Reverse-Engineered Reasoning for Open-Ended Generation"
_ICLR.cc/2026/Conference — ICLR 2026 Poster_

### Official Review · Reviewer_bq1Q · 2025-10-30

**Soundness:** 3
**Presentation:** 3
**Contribution:** 2
**Rating:** 4
**Confidence:** 4

**Summary:**

This paper introduces Reverse-Engineered Reasoning (REER), a novel paradigm that generates reasoning data for open-ended tasks by working backward from known high-quality solutions instead of using costly RL or distillation. This method frames trajectory discovery as a gradient-free search to find the reasoning path that minimizes the solution's perplexity, resulting in a model trained on this data that competes with top proprietary models.

**Strengths:**

1. The motivation is clear and important.
2. The paper is well-written and easy to follow.
3. The proposed method has potential for other domains where reasoning paths are rare and hard to obtain.

**Weaknesses:**

1. The core of the entire method is to assume that minimizing PPL ($y | x, z $) can find a high-quality inference trajectory $ z $. However, the paper does not prove the direct and reliable correlation between PPL and reasoning quality (e.g., efficiency, creativity, and logical depth). PPL often represents the quality and fluency of language, yet it is not equivalent to reasoning quality.
2. The computation of PPL is based on the base model; however, the value of PPL can vary across different LLMs. The paper only conducts experiments on Qwen3-8B, which is insufficient to demonstrate the generalizability in different LLMs.
3. It would be interesting to investigate the performance across different model sizes, which can further verify the ubiquity of the proposed method.
4. The depth and diversity of reasoning paths lack a quantitative evaluation.
5. Although claiming to be more cost-effective than distillation and RL, the paper does not provide a specific cost analysis or comparison of the REER synthesis process, which involves multiple LLM calls and PPL calculations.
6. The evaluation (e.g., PPL) of reasoning path quality is narrow, and it is hard to verify whether the reasoning paths are correct. Wrong reasoning paths can also have low PPL due to other factors (e.g., hallucination), leading to my concern regarding real-world application and applying to other tasks.

**Questions:**

See weaknesses.

---

> ### Author Response · Authors · 2025-11-24
> **Author Rebuttal (Part1)**
>
> # **Response to Reviewer bq1Q**
>
> We sincerely thank the reviewer for the thoughtful evaluation. We appreciate that you found our motivation clear and important, and recognized the potential of our method for domains where reasoning paths are scarce.
>
> We believe the following clarifications and new results will address your concerns, and we wish you could re-evaluate the paper and consider supporting paper accpetance.
>
>
> ### **1. Validity of Perplexity (PPL) as a Proxy (Response to W1, W6)**
>
> **Summary of Concern:** You questioned whether minimizing PPL$(y|x,z)$ reliably correlates with reasoning quality (efficiency, creativity, logical depth), noting that PPL often represents fluency rather than logic.
>
> **Clarification:**
> 1.  **We clarify that the goal of minimizing PPL$(y^\star|x,z)$ is not to measure reasoning quality,** but serves as a proxy of the likelihood of generating a high-quality final writing solution $y^\star$ given a deep reasoning $z$. [cite_start]The PPL is minimized when the deep reasoning $z$ provides a coherent logical bridge from the query $x$ to the final high-quality solution $y^\star$[cite: 27, 59].
> 2.  While PPL$(y^\star|x,z)$ does not measure reasoning quality directly, optimizing this objective (outcome signal on high-quality solution) **implicitly optimizes the reasoning process,** analogous to how Outcome-Supervised Reinforcement Learning (RL) improves reasoning using only final-answer supervision.
>
> Below, we provide a formal derivation to demonstrate the equivalence between our objective, MLE, and standard RL objective with a reward model, followed by an explanation of the implicit optimization mechanism.
>
> ---
>
> **1.1 Formal Derivation**
>
> **Step 1: Connecting Minimizing PPL to Maximizing Likelihood**
> First, we establish the relationship between Perplexity (PPL) and the likelihood of the reference solution. Let $x$ be the query, $z$ be the latent reasoning chain, and $y^\star$ be the known high-quality reference solution. The PPL of $y^\star$ conditioned on the reasoning $z$ is defined as the inverse normalized log-likelihood:
>
> $$
> PPL(y^\star | x, z) = \exp \left( - \frac{1}{N} \log \pi(y^\star | x, z) \right)
> $$
>
>
> From this definition, it is evident that **minimizing PPL is mathematically equivalent to maximizing the likelihood $\pi(y^\star | x, z)$**. In other words, we are searching for a reasoning path $z$ that makes the reference high-quality solution $y^\star$ maximally probable.
>
> **Step 2: Connecting Maximizing Likelihood to Reward Maximization (RL)**
> Next, we show that maximizing this likelihood is equivalent to maximizing the expected reward in a Reinforcement Learning objective.
>
> Consider a standard RL objective $J_{RL}$, where the goal is to maximize the expected reward of generating a response $y$. Let $P(y\in Y^\ast|y)$ denote the likelihood that a generated response $y$ is of high quality. We posit that the reward function $R(y)$ is proportional to this probability:
>
> $$R(y) \propto P(y \in Y^\ast|y)$$
>
> The standard RL objective is the expectation of this reward over the generation policy. Expanding the inner expectation over the space of possible responses $y$, we get:
>
> $\min_\pi \mathbb{E}_{z \sim \pi(\cdot|x)} \left[\sum_y {\pi(y|x, z)} \cdot R(y)  \right] $
>
> $=\min_\pi \mathbb{E}_{z \sim \pi(\cdot|x)} \left[ \pi( y \in Y^\ast |x, z) \right] $
>
> This derivation demonstrates the equivalence of reward maximization and MLE, which also aligns with existing work VeriFree [1].
> This connection implies that, if the reward model is trained on the same high-quality writing solution, it is mathematically equivalent to directly measuring the likelihood or PPL of generating those reference solution.
>
>
> ***
>
> **1.2 REER implicitly pushes reasoning quality like Outcome-Based RL**
>
> Given that minimizing PPL is mathematically equivalent to maximizing the reward of a high-quality outcome, we now explain how this outcome-based signal improves the intermediate reasoning process $z$.
>
> * **Analogy to Outcome-Supervised RL:** In standard reasoning RL (e.g., GRPO), the model receives a binary outcome reward solely based on the final answer correctness. Despite lacking explicit supervision on the *steps* of reasoning, the model learns to generate better reasoning traces because valid logic is the most reliable path to the correct solution.
>
> Like standard outcome-supervised RL, our approach does not explicitly reward reasoning process, but can implicitly improves it using the PPL signal.
>
> The underlying reason is: the pre-trained model has learned the foundamental language structure that values coherence and logic. If a candidate $z$ is hallucinated, irrelevant, or logically flawed, the pre-trained model perceives the transition to the specific high-quality solution $y^\star$ as improbable (high "surprise" or PPL), based on its language prior.
>
> [1] VeriFree: Verifying Open-Ended Generation without a Verifier.

---

> ### Author Response · Authors · 2025-11-24
> **Author Rebuttal (Part 2)**
>
> ### **2. Generalizability Across Models and Sizes (Response to W2, W3)**
> **Core Concern:** You requested evidence of performance on different LLMs and model sizes.
>
> **Clarification:**
> As we mention in the experimental setup, Qwen3-8B is selected for its small size and superior context handling, which is both amenable to the fellow researchers and critical for long reasoning traces. However, to address your concern about ubiquity, we report new results for Qwen2.5-32B on the WritingBench.
>
> | Benchmark Category | Qwen2.5-32B-Instruct | LongWriter-8B | DeepWriter-8B | **DeepWriter-32B** |
> | :--- | :--- | :--- | :--- | :--- |
> | **WB-A** (Academic/Eng) | 52.5 | 57.9 | 72.2 | **73.9** |
> | **WB-B** (Finance/Biz) | 49.8 | 53.9 | 71.8 | **73.3** |
> | **WB-C** (Politics/Law) | 51.0 | 49.0 | 69.8 | **71.4** |
> | **WB-D** (Lit/Arts) | 49.6 | 52.0 | 70.6 | **73.3** |
> | **WB-E** (Education) | 53.9 | 52.9 | 73.7 | **75.9** |
> | **WB-F** (Marketing) | 54.2 | 52.0 | 72.3 | **73.7** |
>
> The results show consistent performance gains like those observed on the 8B model. Specifically, for Lit/Arts, which requires further domain knowledge, there is a significant improvement, aligning with our analysis in the paper. For further improvement, we would likely need to continuously scale up the data diversity and the generator model.
> ***
> ### **3. Cost-Effectiveness and Efficiency Analysis (Response to W5)**
>
> **Summary of Concern:** You requested a specific cost analysis comparing REER to Distillation and RL.
>
> **Response:**
> We have prepared a concrete comparison to illustrate the efficiency of REER. The table below estimates the resources required to generate/train on the 20k dataset for the same query set.
>
> | Method | Training Requirement | Inference Cost | Estimated Time (20k Queries) | Dependency |
> | :--- | :--- | :--- | :--- | :--- |
> | **Distillation** | Low (SFT only) | **Very High** (Proprietary APIs) | ~48x Hours (x considers the iterations on prompt workflows) | Bound by Teacher Capability |
> | **RL (PPO)** | **Very High** (Train RM + Rollouts) | High (Generations during train) | ~120+ Hours (Gradient updates) | Requires Reward Model (RM) |
> | **REER (Ours)** | **Low** (SFT only) | **Moderate** (Local Search) | 4 Hours for SFT and 8 Hours for synthesis | Scalable |
>
> * **Cost Advantage:** Distillation for 20k long-context traces via GPT-4o is estimated to cost $5,000~$50,000 USD. REER generates these using open-source models.
> * **Time Advantage:** Unlike RL, which requires computationally expensive gradient updates and rollout storage during the data creation loop, REER is **gradient-free** during the search phase. It relies purely on inference, allowing for efficient optimization like massive parallelization and KV cache.
> ***
> ### **4. Evaluation of Reasoning Depth (Response to W4)**
> We agree with the reviewer that strictly quantifying "reasoning depth" and reasoning diversity are important. We wish to highlight that, **we have provided relevant analysis in the current manuscript, despite the challenges of perfectly evaluating these measures.**
>
> Ideally, a reasoning trace should be modeled as a semantic graph to measure logical hops and structural complexity. However, how to convert reasoning trace properly to a graph remains itself an unsolved challenge. Direct metrics of token length risk treating reasoning as a linear chain, which inevitably confounds "depth" with "verbosity"—a longer chain is not necessarily deeper.
>
> Despite these issues, we have included the following analysis in the current manuscript to support the evaluation of reasoning depth and diveristy.
>
> 1.  **Distributional Analysis of token length (Figure 9 in Appendix):** We present the token length distribution of our generated traces compared to baselines. While length is not solely equivalent to depth, the substantial shift toward longer, structured outputs -- without the presence of repetitive loops (which our pipeline explicitly filters) -- indicates a higher complexity of thought.
> 2.  **Depth of Analysis (Figure 6):** We conducted a "Depth of Analysis" evaluation using strong model judges (highly correlated with human preference). This metric specifically relates with reasoning depth
> 3. **Diverisity (Figure 7):** In Appendix B.2 Figure 7, we analyzed the diversity of thinking patterns. We compared the frequency of reasoning phrases (e.g., "Wait, but," "Alternatively") between models. The data shows that our method produces a "more diverse and balanced distribution of patterns," whereas the baseline relies on repetitive, formulaic phrases.
>
> ***
> ## **Final Appeal**
> **We kindly invite the reviewer to read our General Response, which summarizes the value of our work and the promising future directions inspired from it. Given the above clarifications and new results, we believe we address your concerns and we respectfully request that you re-evaluate the paper and consider raising your score to support its acceptance.**

---

> > ### Comment · Reviewer_bq1Q · 2025-11-27
> >
> > I appreciate the feedback and resolve most of my concerns. I raise the score to 6.

---

> > > ### Author Response · Authors · 2025-11-27
> > > **Thank You**
> > >
> > > Dear Reviewer bq1Q,
> > >
> > > We are very grateful for your reassessment of our work and for raising your score. Your insightful comments throughout this process have been valuable to us.
> > >
> > > Thank you again for your time and effort. We wish you a smooth path in your own research endeavors.
> > >
> > > Best,
> > > The Authors

---

### Official Review · Reviewer_ngby · 2025-10-31

**Soundness:** 3
**Presentation:** 3
**Contribution:** 3
**Rating:** 6
**Confidence:** 3

**Summary:**

This paper introduces Reverse-Engineered Reasoning (REER), a new paradigm for teaching deep reasoning in open-ended, creative tasks. Traditional methods like RL and distillation fail here due to a lack of clear reward signals or high costs. REER works "backwards": it starts with a high-quality solution and computationally discovers the latent reasoning process that could have produced it. This is framed as a gradient-free search to find a thinking trajectory that minimizes the perplexity of the known-good answer. Using this scalable method, the authors created the DeepWriting-20K dataset. The resulting model, DeepWriter-8B, rivals or exceeds top proprietary models like GPT-40 and Claude 3.5 on challenging writing benchmarks.

**Strengths:**

1. This paper introduces a novel approach that fundamentally shifts how reasoning is taught. Instead of building reasoning forwards through trial-and-error (RL) or imitation (distillation), it works backwards from known good solutions to computationally discover the latent, step-by-step thinking process that could have produced them. This provides a new path for tackling deep reasoning in open-ended, non-verifiable domains.

1. REER is a scalable, gradient-free approach. It successfully bypasses the sample inefficiency of RL and the prohibitively expensive cost of distillation. This creates a cost-effective and automatable pathway to generate vast quantities of high-quality, deep-thinking training data at a scale that was previously impractical.

1. The authors used their method to create and open-source DeepWriting-20K, a large-scale dataset with 20,000 deep reasoning trajectories. This comprehensive resource, spanning 25 categories, is designed to mitigate data scarcity and catalyze future research into planning and structured thought for open-ended generation tasks.

1. The resulting 8B model, DeepWriter-8B, outperforms strong open-source baselines. More significantly, it achieves performance competitive with leading proprietary models like GPT-4o and Claude 3.5. This provides rigorous empirical evidence that human-like deep reasoning can be cultivated without relying on costly distillation or RL.

**Weaknesses:**

1. The paper's main evaluations rely on using powerful LLMs (like GPT-4o and Claude 3.7) as judges. The authors acknowledge that this method, while scalable, has potential for inherent biases.

1. The core REER algorithm depends on using the perplexity (PPL) of the reference answer as a proxy for the quality of the reasoning trajectory. The assumption that lower PPL, which makes the answer seem maximally probable and logical, directly equates to a higher-quality reasoning process is a proxy and may not perfectly align with human judgments of logic, creativity, or efficiency.

1. The primary experiments were conducted on an 8B base model (Qwen3-8B-Base). The effectiveness of REER on much larger models is not yet validated, as mentioned by the authors in Section 4.

**Questions:**

1. What if we use a reward model (e.g., a process reward model) as a proxy for the quality of the reasoning trajectory? Will REER yield better performance?

---

> ### Author Response · Authors · 2025-11-24
> **Author Rebuttal**
>
> # **Response to Reviewer ngby**
>
> We sincerely thank the reviewer for the positive assessment and for recognizing REER as a "novel paradigm" that offers a "cost-effective and automatable pathway" for deep reasoning. We are encouraged that you found our dataset, DeepWriting-20K, to be a valuable resource for the community.
>
> Below, we address your questions regarding reward modeling and model scalability, and discuss the evaluation limitations.
>
> ### **1. Process Reward Models vs. Perplexity (Response to Q1)**
>
> **Question:** "What if we use a reward model (e.g., a process reward model) as a proxy for the quality of the reasoning trajectory? Will REER yield better performance?"
>
> **Response:**
> This is an insightful suggestion. To clarify our current approach: we use PPL as a proxy for the conditional generation likelihood $P(y^\star | x, z)$. In essence, we search for the reasoning path $z$ that makes the known high-quality solution $y^\star$ maximally probable. As detailed in our response to other reviewers, minimizing this PPL is mathematically equivalent to maximizing the likelihood of the reference solution, which implicitly optimizes the logical coherence of the bridge between query and answer.
>
> **We strongly agree** that incorporating a Process Reward Model (PRM) to explicitly guide the quality of intermediate reasoning steps is a promising direction and will yield further performance gains because it achieves better quality control over the synthesized reasoning trajectories. However, we note a key challenge: unlike math or code, where process verification is objective, building a reliable PRM for "creative reasoning" is notoriously difficult and currently an open research problem. **We appreciate you pointing out this future direction** and will put this to future work
>
> ### **2. Scalability to Larger Models (Response to W3)**
> We acknowledge this concern. To demonstrate the scalability of our approach, **we have conducted new experiments applying our methodology to the larger **Qwen2.5-32B** model.** The results on WritingBench (WB) are presented below:
>
> | Benchmark Category | Qwen2.5-32B-Instruct | LongWriter-8B | DeepWriter-8B | **DeepWriter-32B** |
> | :--- | :--- | :--- | :--- | :--- |
> | **WB-A** (Academic/Eng) | 52.5 | 57.9 | 72.2 | **73.9** |
> | **WB-B** (Finance/Biz) | 49.8 | 53.9 | 71.8 | **73.3** |
> | **WB-C** (Politics/Law) | 51.0 | 49.0 | 69.8 | **71.4** |
> | **WB-D** (Lit/Arts) | 49.6 | 52.0 | 70.6 | **73.3** |
> | **WB-E** (Education) | 53.9 | 52.9 | 73.7 | **75.9** |
> | **WB-F** (Marketing) | 54.2 | 52.0 | 72.3 | **73.7** |
>
> The results show that DeepWriter-32B achieves consistent performance gains over the strong base model (Qwen2.5-32B-Instruct), particularly in domains requiring domain-specific knowledge (e.g., Literature & Arts). This confirms that REER scales effectively to larger parameter counts. Despite the relative improvement against the base model, we observed that there appears an ceiling for further absolute performance improvement (e.g., the improvement is saturated on HelloBench and LongBench and somewhat marginal on WritingBench). For further improvement, we would likely need to continuously scale up the data diversity and the generator model.
>
> ### **3. Reliance on LLM Judges (Response to W1)**
>
> We fully acknowledge this limitation. As the reviewer noted, open-ended generation lacks a singular ground truth, making evaluation inherently difficult. We adopted the LLM-as-a-judge protocol because it is the current standard in the field (used by benchmarks like HelloBench and WritingBench) and the only scalable way to evaluate 20,000+ long-context samples.
>
> To mitigate potential bias, we employed multiple distinct judges (Claude-3.7 and GPT-4o) across different benchmarks. While we believe our large performance margins (e.g., +18 points on WritingBench) are robust to judge noise, we agree that developing better evaluation for open-ended tasks is a critical next step for the field, which we intend to address in future work.
> ***
> ## **Final Appeal**
> **We kindly invite the reviewer to read our General Response, which summarizes the value of our work and the promising future directions inspired from it. Given the above clarifications and new results, we believe we address your concerns and we respectfully wish you could consider raising your score to support its acceptance.**

---

### Official Review · Reviewer_XQFR · 2025-11-07

**Soundness:** 3
**Presentation:** 3
**Contribution:** 3
**Rating:** 8
**Confidence:** 3

**Summary:**

This paper introduces REverse-Engineered Reasoning (REER), a new paradigm that infers plausible step-by-step reasoning paths backwards from high-quality outputs, instead of constructing them via reinforcement learning or distillation. By formulating reasoning-path recovery as a gradient-free local search minimizing perplexity, the authors synthesize 20 K structured reasoning trajectories (DeepWriting-20K) and fine-tune an 8B model (DeepWriter-8B). The model achieves near-proprietary-level performance on LongBench-Write, HelloBench, and WritingBench, outperforming all open-source baselines.
Overall, the work provides a cost-effective way to instill deep reasoning for open-ended generation. It offers both a scalable data-generation framework and strong empirical evidence that reverse-engineered reasoning can rival large proprietary models without RL or teacher supervision.

**Strengths:**

1. Scalable and Cost-Effective Framework: REER eliminates the need for teacher models or reward functions by using a gradient-free local search guided by perplexity, enabling large-scale, low-cost synthesis of deep reasoning trajectories.
2. High-Quality Public Dataset: The released DeepWriting-20K dataset of 20 K structured reasoning trajectories across 25 domains is a valuable open resource for future research on reasoning and creative generation.
3. Strong Empirical Results: The proposed DeepWriter-8B model achieves near-proprietary-level performance on LongBench, HelloBench, and WritingBench, significantly outperforming all open-source baselines.
4. Extending Deep Reasoning to Non-Verifiable Tasks: REER successfully brings deep reasoning to open-ended and creative domains (e.g., writing), bridging the gap between verifiable reasoning (math, code) and subjective generation tasks.

**Weaknesses:**

1. Subjective Evaluation via LLM Judges: Benchmarks rely heavily on GPT-4o or Claude-3.7 as evaluators, which introduces bias and limits reproducibility; no human or standardized blind evaluation is included.
2. Scalability and Efficiency Trade-offs: The iterative local search for trajectory refinement can be computationally expensive for large datasets, and its efficiency compared to distillation or RL remains under-analyzed.

**Questions:**

1. Under what conditions does REER fail to find meaningful reasoning trajectories (e.g., nonsensical outputs, repetitive loops), and how are such cases handled?
2. How would REER behave when applied to larger base models or multimodal datasets—would it continue to scale effectively or face diminishing returns?

---

> ### Author Response · Authors · 2025-11-24
> **Author Response**
>
> # **Response to Reviewer XQFR**
>
> **We are deeply encouraged by your positive assessment of our work.** We are glad that you recognized the value of our REverse-Engineered Reasoning (REER) paradigm as a scalable and cost-effective framework, as well as the significance of the DeepWriting-20K dataset for the community.
>
> Below, we address your questions regarding failure modes and scalability, and provide the efficiency analysis you requested.
>
> ### **1. Efficiency and Scalability Analysis (Response to W2)**
> **We provide a concrete comparison to illustrate the efficiency of REER.** The table below estimates the resources required to generate/train on the 20k dataset for the same query set.
>
> | Method | Training Requirement | Inference Cost | Estimated Time (20k Queries) | Dependency |
> | :--- | :--- | :--- | :--- | :--- |
> | **Distillation** | Low (SFT only) | **Very High** (Proprietary APIs) | ~48x Hours (x considers workflows) | Bound by Teacher Capability |
> | **RL (PPO)** | **Very High** (Train RM + Rollouts) | High (Generations during train) | ~120+ Hours (Gradient updates) | Requires Reward Model (RM) |
> | **REER (Ours)** | **Low** (SFT only) | **Moderate** (Local Search) | 4 Hours (SFT) + 8 Hours (Synthesis) | Scalable |
>
> **Key Takeaways:**
> * **Gradient-Free Efficiency:** Unlike RL, which requires computationally expensive gradient updates and rollout storage during the data creation loop, REER is **gradient-free** during the search phase. It relies purely on inference, allowing for optimizations like massive parallelization and KV caching.
> * **Cost Advantage:** Distillation for 20k long-context traces via GPT-4o is estimated to cost \$5,000–\$50,000 USD. REER generates these using open-source models at a fraction of the cost.
>
> ### **2. Failure Modes and Quality Control (Response to Q1)**
>
> **We analyzed the raw outputs of the search process and identified a specific failure mode in a small minority of cases.**
>
> In these instances, the reasoning trace may include repetitive sentences or "stalling" behaviors (e.g., repeating a plan multiple times without adding detail). Interestingly, because these traces eventually transition to the correct high-quality reference solution, the Perplexity (PPL) of the reference solution remains low. The logic "bridge" is technically valid—the model is not "surprised" by the answer given the context—but the reasoning path is inefficient.
>
> **Handling Strategy:**
> While these traces satisfy the PPL objective, they are detrimental to training as they can lead to degenerate policy behaviors (e.g., the model learning to pad reasoning to increase length). To handle this, we implement **Quality Control** via heuristic filtering:
> 1.  **Repetition Filtering:** We prune trajectories with high n-gram repetition rates.
> 2.  **End-of-Thinking Filtering:** We remove samples where "planning" language persists into the final 10% of the trace, ensuring the model learns to conclude reasoning before generating the answer.
>
> ### **3. Scaling to Larger Models (Response to Q2)**
> **To address this, we conducted additional experiments applying our dataset and training methodology to the larger Qwen2.5-32B model.** The results on WritingBench (WB) demonstrate that the benefits of REER scale effectively.
>
> | Benchmark Category | Qwen2.5-32B-Instruct | LongWriter-8B | DeepWriter-8B | **DeepWriter-32B** |
> | :--- | :--- | :--- | :--- | :--- |
> | **WB-A** (Academic/Eng) | 52.5 | 57.9 | 72.2 | **73.9** |
> | **WB-B** (Finance/Biz) | 49.8 | 53.9 | 71.8 | **73.3** |
> | **WB-C** (Politics/Law) | 51.0 | 49.0 | 69.8 | **71.4** |
> | **WB-D** (Lit/Arts) | 49.6 | 52.0 | 70.6 | **73.3** |
> | **WB-E** (Education) | 53.9 | 52.9 | 73.7 | **75.9** |
> | **WB-F** (Marketing) | 54.2 | 52.0 | 72.3 | **73.7** |
>
> DeepWriter-32B achieves consistent improvements over its base instruct version and DeepWriter-8B, particularly in domains requiring extensive knowledge (e.g., Literature & Arts). This suggests that as the base model's capacity increases, REER can unlock even deeper reasoning and domain-specific planning capabilities. Despite the relative improvement against the base model, we observed that there appears an ceiling for further absolute performance improvement (e.g., the improvement is saturated on HelloBench and LongBench and somewhat marginal on WritingBench). For further improvement, we would likely need to continuously scale up the data diversity and the generator model.
>
> ***
> ### **Final Appeal**
> **We kindly invite the reviewer to read our General Response, which summarizes the value of our work and the promising future directions inspired from it. Given the above clarifications and new results, we believe we address your concerns and we respectfully wish you could keep your score to support its acceptance.**

---

### Author Response · Authors · 2025-11-24
**General Response**

# General Response to Area Chairs and Reviewers

We sincerely thank the Area Chairs and all reviewers for their time and insightful feedback. We are encouraged that the reviewers unanimously recognized the novelty of our approach and the value of our contributions to the community.

### **Highlights of Reviewer Feedback**
We appreciate that the reviewers highlighted the following key strengths of our work:
* **Value of REER:** Reviewers commended REER as a "novel paradigm" that "fundamentally shifts how reasoning is taught" (Reviewer ngby), highlighted our method as a "scalable and cost-effective framework" (Reviewer XQFR), and noted its great "potential for other domains where reasoning paths are rare and hard to obtain" (Reviewer bq1Q).
* **Strong Empirical Results:** All reviewers acknowledged the strong performance. Reviewer ngby noted that DeepWriter-8B "rivals or exceeds top proprietary models like GPT-4o," and Reviewer XQFR confirmed it "significantly outperforms all open-source baselines."
* **Community Contribution:** Reviewers XQFR and bq1Q emphasized the value of **DeepWriting-20K** as a "high-quality public dataset" and a "valuable open resource" for domains where reasoning data is scarce.
* **Clear Presentation:** Reviewers praised the paper for being clear, well-written, and easy to follow, with high presentation scores across the board.

### **Summary of Rebuttal to Common Concerns**
We have addressed the primary concerns raised by the reviewers with detailed clarifications and new experiments:

* **Validity of Perplexity (PPL) as a Proxy (Reviewers bq1Q, ngby):**
    We clarified that we do not use PPL to measure reasoning quality directly in a vacuum. Instead, minimizing the PPL of a *known high-quality reference solution* ($y^\star$) is mathematically equivalent to maximizing the likelihood of the optimal outcome given the reasoning. This acts as an outcome-supervised signal that implicitly forces the model to learn a coherent, logical reasoning path ($z$) most probable bridge to the high-quality solution.

* **Scalability to Larger Models (Reviewers bq1Q, ngby, XQFR):**
    To demonstrate ubiquity, we conducted new experiments applying REER to **Qwen2.5-32B**. The results confirm that our method scales effectively: DeepWriter-32B achieves consistent performance gains over the strong base model and further surpasses the 8B version, particularly in domains requiring extensive literature knowledge (e.g., Literature & Arts).

* **Cost-Effectiveness (Reviewers bq1Q, XQFR):**
    We provided a concrete cost analysis in our individual responses. Because REER is **gradient-free** during the search phase and relies on local inference (leveraging optimizations like KV caching), it is significantly more efficient than Reinforcement Learning (which requires computationally expensive RL training, rollouts, and Reward Model training) and orders of magnitude cheaper than distillation via proprietary APIs.
***
### **Promising Future Directions**
This work bridges a critical gap in the field: enabling "deep reasoning" for open-ended, creative tasks where ground-truth verification is challenging. We envision meaningful research based on our work including but not limited to
* **Scaling to other open-ended/costly-to-verify tasks:** We see immense potential in adopting REER for other tasks, such as agentic software engineering, GUI agents, which involves great engineering complexity to interact with docker environments.
* **Data for Pre-training/SFT:** REER can serve as a data engine to generate **logically coherent synthetic data** to enrich SFT or pre-training corpora, further unleashing model potential.

**We are also delighted to share that the reviewers constructive feedback has also inspired promising ideas of improving quality control of deep reasoning synthesis using process verifiers (ngby, bq1Q).**

***
### **Final Appeal**
We believe this work contribute more than a method to the community; it is
- **scalable:** We introduced REverse-Engineered Reasoning (REER), a scalable paradigm that operationalizes the "reverse-engineering" of latent thought processes, fundamentally shifting the approach to reasoning away from the dependencies of costly reinforcement learning or teacher distillation
- **validated:**: We demonstrated using both 8B and 32B models that sophisticated planning and deep reasoning are attainable in open-source models without prohibitive computational budgets
- **Democratized:** By curating and open-sourcing the first large-scale dataset of deep reasoning trajectories for open-ended task --we have dismantled a critical barrier to entry, enabling the broader research community to investigate and build upon high-quality, structured thought data

**Given the novelty of the REER paradigm and the significant contribution of our work to the research community, we respectfully ask the reviewers to support the acceptance of this paper.**

---

### Author Response · Authors · 2025-12-01
**Summary of Author-Discussion Results**

Dear ACs and Reviewers,

We would like to express our sincere gratitude for the time and effort you dedicated to reviewing our paper. Your comments have been instrumental in improving the rigor and clarity of our work.

During the discussion period, we carefully addressed the concerns of all reviewers, as summarized in the General Response below. **Specifically, we are pleased that reviewer XQFR (score 8) and ngby (score 6) expressed positive recognition of our work. After our rebuttal, reviewer bq1Q also acknowledged our clarifications and accordingly provided positive rating (score 6).**

Thank you all once again for your substantial efforts throughout the review and discussion process.

Best regards,

Authors

---

### Meta-Review · Area_Chair_LZsG · 2026-01-07

**Summary:**

This work brings REverse-Engineered Reasoning (REER), a novel paradigm that infers plausible step-by-step reasoning paths backwards from high-quality outputs, rather than constructing them via reinforcement learning or distillation. The REER is a scalable and gradient-free method. It successfully bypasses the sample inefficiency of RL and the prohibitively expensive cost of distillation. The paper is well-written. The proposed approach has potential for other domains where reasoning paths are rare and hard to obtain.

**Reviewer Concerns:**

The major concerns seem to be well replied.

**Reviewer Scores:**

Reviewers likely will keep the rating.

---

### Decision · Program_Chairs · 2026-01-26

Accept (Poster)